# Low-Budget Active Learning via Wasserstein Distance: An Integer Programming Approach

**Rafid Mahmood**[1]     **Sanja Fidler**[1,2,3]     **Marc T. Law**[1]
[1]NVIDIA     [2]University of Toronto     [3]Vector Institute
{rmahmood, sfidler, marcl}@nvidia.com

## Abstract

Active learning is the process of training a model with limited labeled data by selecting a core subset of an unlabeled data pool to label. The large scale of data sets used in deep learning forces most sample selection strategies to employ efficient heuristics. This paper introduces an integer optimization problem for selecting a core set that minimizes the discrete Wasserstein distance from the unlabeled pool. We demonstrate that this problem can be tractably solved with a Generalized Benders Decomposition algorithm. Our strategy uses high-quality latent features that can be obtained by unsupervised learning on the unlabeled pool. Numerical results on several data sets show that our optimization approach is competitive with baselines and particularly outperforms them in the low budget regime where less than one percent of the data set is labeled.

## 1 Introduction

Although deep learning demonstrates superb accuracy on supervised learning tasks, most deep learning methods require massive data sets that can be expensive to label. To address this challenge, active learning is a labeling-efficient supervised learning paradigm when given a large unlabeled data set (Cohn et al., 1994; Settles, 2009). Active learning uses a finite budget to select and label a subset of a data pool for downstream supervised learning. By iteratively training a model on a current labeled set and labeling new points, this paradigm yields models that use a fraction of the data to achieve nearly the same accuracy as classifiers with unlimited labeling budgets.

The task of selecting points to label is a combinatorial optimization problem of determining the most *representative* subset of the unlabeled data (Sener & Savarese, 2017). However, to estimate the value of labeling a point, active learning strategies require informative features from the unlabeled data. Integer optimization models can also grow impractically quickly with the data set size. Nonetheless, optimally selecting which points to label can reduce operational costs, especially for applications such as medical imaging where labeling requires time-consuming human expert labor (Rajchl et al., 2016), or domain adaptation where data pools are hard to obtain (Saenko et al., 2010).

In this paper, we introduce an optimization framework for active learning with small labeling budgets. To select points, we minimize the Wasserstein distance between the unlabeled set and the set to be labeled. We prove that this distance bounds the difference between training with a finite versus an unlimited labeling budget. Our optimization problem admits a Generalized Benders Decomposition solution algorithm wherein we instead solve sub-problems that are orders of magnitude smaller (Geoffrion, 1972). Our algorithm guarantees convergence to a globally optimal solution and can be further accelerated with customized constraints. Finally, to compute informative features, we use unsupervised learning on the unlabeled pool. We evaluate our framework on four visual data sets with different feature learning protocols and show strong performance versus existing selection methods. In particular, we outperform baselines in the very low budget regime by a large margin.

Our overall contributions are as follows: (1) We derive a new deterministic bound on the difference in training loss between using the full data set versus a subset via the discrete Wasserstein distance. This further bounds the overall expected risk. (2) We propose active learning by minimizing the Wasserstein distance. We develop a globally convergent customized algorithm for this problem using Generalized Benders Decomposition. (3) We consider low budget active learning for image classification and domain adaptation where up to 400 images can be labeled and show that optimally selecting the points to label in this regime improves over heuristic-based methods.

Figure 1: (Left) In our active learning framework, we first pre-train our features with self-supervised learning, select samples to label by minimizing the discrete Wasserstein distance, and then train our classifier. (Right) $t$-SNE plot of the feature space on the STL-10 data set with selected points highlighted. The baseline misses the top right region. See Appendix E.7 for full visual comparisons.

## 2 PROPOSED MODEL

We first set up the active learning problem and develop the main theory motivating our active learning strategy illustrated in Figure 1. All of our proofs can be found in Appendix A.

### 2.1 PRELIMINARIES OF ACTIVE LEARNING

Consider a $C$-class classification problem over features $\mathbf{x} \in \mathcal{X}$ and labels $y \in \mathcal{Y} := \{1, \ldots, C\}$, where $(\mathcal{X}, \|\cdot\|)$ and $(\mathcal{Y}, |\cdot|)$ are metric spaces. With a data distribution and a loss function $\ell(\mathbf{x}, y; \mathbf{w}) : \mathcal{X} \times \mathcal{Y} \to \mathbb{R}$ parametrized by $\mathbf{w}$, our goal is to minimize the expected risk $\min_{\mathbf{w}} \mathbb{E}[\ell(\mathbf{x}, y; \mathbf{w})]$. In practice, we use a labeled data set $\{(\mathbf{x}_i, y_i)\}_{i=1}^N$ of $N$ samples to minimize the empirical risk.

In active learning, we instead have features $\mathcal{D} = \{\mathbf{x}_i\}_{i=1}^N$ and a labeling oracle $\Omega : \mathcal{X} \to \mathcal{Y}$. Let $\boldsymbol{\pi} \in \{0, 1\}^N$ indicate points to label given a budget $B$. That is, we may call the oracle $B$ times to create a labeled core set $\mathcal{C}(\boldsymbol{\pi}) := \{(\mathbf{x}_j, \Omega(\mathbf{x}_j)\}_{j=1}^B$ where $\mathbf{x}_i \in \mathcal{C}(\boldsymbol{\pi})$ is labeled iff $\pi_i = 1$. An active learning strategy is an algorithm to optimize $\mathcal{C}(\boldsymbol{\pi})$. We omit $\boldsymbol{\pi}$ in the notation for $\mathcal{C}$ when obvious.

Active learning strategies can be categorized into uncertainty-based, representation-based, or hybrid. Uncertainty methods determine images to label by using measures of the uncertainty of a classifier (Roth & Small, 2006; Li & Guo, 2013; Wang & Shang, 2014). Representation methods select a diverse core set of examples spread over the feature space (Yu et al., 2006; Sener & Savarese, 2017; Contardo et al., 2017). Uncertainty estimates and latent features are commonly obtained by training a preliminary classifier for feature extraction with a previously labeled data set (Sinha et al., 2019; Shui et al., 2020). Other approaches also include pre-clustering (Nguyen & Smeulders, 2004) or few-shot learning (Woodward & Finn, 2017).

Sener & Savarese (2017) bound the expected risk by a generalization error of training with an unlimited number of oracle calls, the empirical risk of training with a core set of $B$ points, and a core set loss for training with all points versus only the core set. This bound, slightly revised, is

$$\mathbb{E}[\ell(\mathbf{x}, y; \mathbf{w})] \leq \underbrace{\frac{1}{B} \sum_{j=1}^B \ell(\mathbf{x}_j, \Omega(\mathbf{x}_j); \mathbf{w})}_{\text{empirical risk}} + \underbrace{\left| \mathbb{E}[\ell(\mathbf{x}, y; \mathbf{w})] - \frac{1}{N} \sum_{i=1}^N \ell(\mathbf{x}_i, \Omega(\mathbf{x}_i); \mathbf{w}) \right|}_{\text{generalization bound}}$$

$$+ \underbrace{\frac{1}{N} \sum_{i=1}^N \ell(\mathbf{x}_i, \Omega(\mathbf{x}_i); \mathbf{w}) - \frac{1}{B} \sum_{j=1}^B \ell(\mathbf{x}_j, \Omega(\mathbf{x}_j); \mathbf{w})}_{\text{core set loss}}.$$

The generalization bound does not depend on the core set. Furthermore for small $B$, the empirical risk is always negligibly small. Thus from Sener & Savarese (2017), we can focus on an active learning strategy that specifically minimizes the core set loss.

### 2.2 ACTIVE LEARNING VIA WASSERSTEIN DISTANCE

As a function of $\mathcal{C}$, the core set loss resembles a distribution matching loss on the empirical risk with $\mathcal{C}$ versus $\mathcal{D}$. This motivates an active learning strategy of minimizing a distance between

distributions over these two sets (Shui et al., 2020). We consider minimizing the discrete Wasserstein distance (Villani, 2008). Let $\mathbf{D_x} = [\|\mathbf{x}_i - \mathbf{x}_{i'}\|]_{i=1,i'=1}^{N,N}$ be a distance matrix over features in $\mathcal{D}$. The discrete Wasserstein distance between $\mathcal{C}$ and $\mathcal{D}$ is denoted by $W(\mathcal{C}(\boldsymbol{\pi}), \mathcal{D})$ and defined as

$$W(\mathcal{C}(\boldsymbol{\pi}), \mathcal{D}) := \min_{\boldsymbol{\Gamma} \geq \mathbf{0}} \left\{ \langle \mathbf{D_x}, \boldsymbol{\Gamma} \rangle \; \middle| \; \boldsymbol{\Gamma}\mathbf{1} = \frac{1}{N}\mathbf{1}, \; \boldsymbol{\Gamma}^\mathsf{T}\mathbf{1} = \frac{1}{B}\boldsymbol{\pi} \right\} \tag{1}$$

$$= \max_{\boldsymbol{\lambda}, \boldsymbol{\mu}} \left\{ \frac{1}{N}\boldsymbol{\mu}^\mathsf{T}\mathbf{1} - \frac{1}{B}\boldsymbol{\lambda}^\mathsf{T}\boldsymbol{\pi} \; \middle| \; \boldsymbol{\lambda}^\mathsf{T} \otimes \mathbf{1} + \boldsymbol{\mu} \otimes \mathbf{1}^\mathsf{T} \leq \mathbf{D_x} \right\} \tag{2}$$

where $\langle \mathbf{A}, \mathbf{B} \rangle := \text{Trace}(\mathbf{A}^\top \mathbf{B})$ is the Frobenius inner product between real matrices $\mathbf{A}$ and $\mathbf{B}$. We then formulate our active learning strategy as the following optimization problem:

$$\min_{\boldsymbol{\pi} \in \{0,1\}^N} W(\mathcal{C}(\boldsymbol{\pi}), \mathcal{D}) \qquad \text{s.t.} \; |\mathcal{C}(\boldsymbol{\pi})| = B. \tag{3}$$

We may also consider alternative distance functions such as divergences. However, since the Wasserstein distance minimizes a transport between distributions, it can lead to better coverage of the unlabeled set (e.g., Shui et al. (2020) demonstrate numerical examples). Furthermore, we compute the discrete Wasserstein distance by a linear program, meaning that our active learning strategy can be written as a mixed integer linear program (MILP). Finally, the Wasserstein distance induces a new bound on the expected risk in training. Specifically, we show below that the Wasserstein distance directly upper bounds the core set loss under the assumption of a Lipschitz loss function.

**Theorem 1.** *Fix $\mathbf{w}$ constant. For any $\varepsilon > 0$, if $\ell(\mathbf{x}, y; \mathbf{w})$ is $K$-Lipschitz continuous over $(\mathcal{X} \times \mathcal{Y}, \|\mathbf{x}\| + \varepsilon|y|)$, then $\frac{1}{N}\sum_{i=1}^N \ell(\mathbf{x}_i, \Omega(\mathbf{x}_i); \mathbf{w}) - \frac{1}{B}\sum_{j=1}^B \ell(\mathbf{x}_j, \Omega(\mathbf{x}_j); \mathbf{w}) \leq KW(\mathcal{C}, \mathcal{D}) + \varepsilon KC.$*

The second term in the bound vanishes for small $\varepsilon$. Moreover, Theorem 1 assumes Lipschitz continuity of the model for fixed parameters, which is standard in the active learning literature (Sener & Savarese, 2017) and relatively mild. We provide a detailed discussion on the assumption in Appendix B. Finally, the bound in Theorem 1 directly substitutes into the bound of Sener & Savarese (2017), meaning that minimizing the Wasserstein distance intuitively bounds the expected risk.

## 3 MINIMIZING WASSERSTEIN DISTANCE VIA INTEGER PROGRAMMING

We now present our Generalized Benders Decomposition (GBD) algorithm for solving problem (3). GBD is an iterative framework for large-scale non-convex constrained optimization (Geoffrion, 1972). We provide a general review of GBD in Appendix C and refer to Rahmaniani et al. (2017) for a recent survey. In this section, we first reformulate problem (3), summarize the main steps of our algorithm, and show three desirable properties: (i) it has an intuitive interpretation of iteratively using sub-gradients of the Wasserstein distance as constraints; (ii) it significantly reduces the size of the original optimization problem; and (iii) it converges to an optimal solution. Finally, we propose several customization techniques to accelerate our GBD algorithm.

### 3.1 MINIMIZING WASSERSTEIN DISTANCE WITH GBD

Problem (3), which selects an optimal core set, can be re-written as the following MILP:

$$\min_{\boldsymbol{\pi} \in \{0,1\}^N, \boldsymbol{\Gamma} \geq \mathbf{0}} \langle \mathbf{D_x}, \boldsymbol{\Gamma} \rangle \qquad \text{s.t.} \;\; \boldsymbol{\Gamma}\mathbf{1} = \frac{1}{N}\mathbf{1} \;, \; \boldsymbol{\Gamma}^\mathsf{T}\mathbf{1} = \frac{1}{B}\boldsymbol{\pi} \;, \; \boldsymbol{\pi}^\mathsf{T}\mathbf{1} = B \tag{4}$$

Let us define $\mathcal{G} := \{\boldsymbol{\Gamma} \geq \mathbf{0} \mid \boldsymbol{\Gamma}\mathbf{1} = 1/N\}$ and $\mathcal{P} := \{\boldsymbol{\pi} \in \{0,1\}^N \mid \boldsymbol{\pi}^\mathsf{T}\mathbf{1} = B\}$. Problem (4) is equivalent to the following Wasserstein-Master Problem (W-MP):

$$\min_{\eta, \boldsymbol{\pi} \in \mathcal{P}} \eta \qquad \text{s.t.} \;\; \eta \geq \inf_{\boldsymbol{\Gamma} \in \mathcal{G}} \left\{ \langle \mathbf{D_x}, \boldsymbol{\Gamma} \rangle + \boldsymbol{\lambda}^\mathsf{T}\left( \frac{1}{B}\boldsymbol{\pi} - \boldsymbol{\Gamma}^\mathsf{T}\mathbf{1} \right) \right\}, \; \forall \boldsymbol{\lambda} \in \mathbb{R}^N \tag{5}$$

Although W-MP contains an infinite number of constraints controlled by the dual variable $\boldsymbol{\lambda}$, it can also be re-written as a single constraint by replacing the right-hand-side with the Lagrangian $L(\boldsymbol{\pi})$ of the inner optimization problem. Furthermore, $L(\boldsymbol{\pi})$ is equivalent to the Wasserstein distance:

$$L(\boldsymbol{\pi}) := \sup_{\boldsymbol{\lambda} \in \mathbb{R}^N} \inf_{\boldsymbol{\Gamma} \in \mathcal{G}} \left\{ \langle \mathbf{D_x}, \boldsymbol{\Gamma} \rangle + \boldsymbol{\lambda}^\mathsf{T}\left( \frac{1}{B}\boldsymbol{\pi} - \boldsymbol{\Gamma}^\mathsf{T}\mathbf{1} \right) \right\} = W(\mathcal{C}(\boldsymbol{\pi}), \mathcal{D}).$$

**Main Steps of the Algorithm.** Instead of the semi-infinite problem W-MP (5), we consider a finite set of constraints $\Lambda \subset \mathbb{R}^N$ and solve a Wasserstein-Relaxed Master Problem (W-RMP($\Lambda$)):

$$\min_{\eta, \boldsymbol{\pi} \in \mathcal{P}} \eta \qquad \text{s.t.} \quad \eta \geq \inf_{\boldsymbol{\Gamma} \in \mathcal{G}} \left\{ \langle \mathbf{D_x}, \boldsymbol{\Gamma} \rangle + \hat{\boldsymbol{\lambda}}^{\mathsf{T}} \left( \frac{1}{B} \boldsymbol{\pi} - \boldsymbol{\Gamma}^{\mathsf{T}} \mathbf{1} \right) \right\}, \; \forall \hat{\boldsymbol{\lambda}} \in \Lambda.$$

Let $(\hat{\eta}, \hat{\boldsymbol{\pi}})$ be an optimal solution to W-RMP($\Lambda$) and let $(\eta^*, \boldsymbol{\pi}^*)$ be an optimal solution to W-MP. Since W-RMP is a relaxation, $\hat{\eta} \leq \eta^*$ lower bounds the optimal value to W-MP (and thus also to problem (4)). Furthermore, $L(\hat{\boldsymbol{\pi}}) = W(\mathcal{C}(\hat{\boldsymbol{\pi}}), \mathcal{D}) \geq W(\mathcal{C}(\boldsymbol{\pi}^*), \mathcal{D}) = \eta^*$ upper bounds the optimal value to W-MP. By iteratively adding new constraints to $\Lambda$, we can make W-RMP($\Lambda$) a tighter approximation of W-MP, and consequently tighten the upper and lower bounds.

In GBD, we initialize a selection $\hat{\boldsymbol{\pi}}^0 \in \mathcal{P}$ and $\Lambda = \emptyset$. We repeat at each iteration $t \in \{0, 1, 2, \dots\}$:

1. Given a solution $(\hat{\eta}^t, \hat{\boldsymbol{\pi}}^t)$, solve the Lagrangian $L(\hat{\boldsymbol{\pi}}^t)$ to obtain a primal-dual transport $(\hat{\boldsymbol{\Gamma}}^t, \hat{\boldsymbol{\lambda}}^t)$.

2. Update $\Lambda \leftarrow \Lambda \cup \{\hat{\boldsymbol{\lambda}}^t\}$ and solve W-RMP($\Lambda$) to obtain a solution $(\hat{\eta}^{t+1}, \hat{\boldsymbol{\pi}}^{t+1})$.

We terminate when $W(\mathcal{C}(\hat{\boldsymbol{\pi}}^t), \mathcal{D}) - \hat{\eta}^t$ is smaller than a tolerance threshold $\varepsilon > 0$, or up to a runtime limit. We omit $t$ in the notation when it is obvious.

**Interpretation.** The Lagrangian is equivalent to computing a Wasserstein distance. Furthermore, the dual variables of the Wasserstein equality constraints are sub-gradients of problem (4) (Rahmaniani et al., 2017). In each iteration, we compute $W(\mathcal{C}(\hat{\boldsymbol{\pi}}), \mathcal{D})$ and add a new constraint to W-RMP($\Lambda$). Intuitively, each constraint is a new lower bound on $\eta$ using the sub-gradient with respect to $\hat{\boldsymbol{\pi}}$.

**Solving Smaller Problems.** The original problem (4) contains $N^2 + N$ variables and $2N + 1$ constraints where $N = |\mathcal{D}|$, making it intractable for most deep learning data sets (e.g., $N = 50,000$ for CIFAR-10). W-RMP($\Lambda$) only contains $N + 1$ variables and $|\Lambda|$ constraints (i.e., the number of constraints equals the number of iterations). In our experiments, we run for approximately 500 iterations, meaning W-RMP($\Lambda$) is orders of magnitude smaller than (4). Although we must also solve the Lagrangian sub-problem in each iteration, computing Wasserstein distances is a well-studied problem with efficient algorithms and approximations (Bonneel et al., 2011; Cuturi, 2013). We discuss our overall reduced complexity in more detail in Appendix D.3.

**Convergence.** Finally, because the objectives and constraints of W-MP are linear and separable in $\boldsymbol{\Gamma}$ and $\boldsymbol{\pi}$ (see Appendix C), GBD converges to an optimal solution in finite time.

**Corollary 1.** *Let $\Lambda^0 = \emptyset$ and $\hat{\boldsymbol{\pi}}^0 \in \mathcal{P}$. Suppose in each iteration $t \in \{0, 1, 2, \dots\}$, we compute $W(\mathcal{C}(\hat{\boldsymbol{\pi}}^t), \mathcal{D})$ to obtain a dual variable $\hat{\boldsymbol{\lambda}}^t$ and compute W-RMP($\Lambda^{t+1} \leftarrow \Lambda^t \cup \{\hat{\boldsymbol{\lambda}}^t\}$) to obtain $(\hat{\eta}^{t+1}, \hat{\boldsymbol{\pi}}^{t+1})$. Then for any $\varepsilon > 0$, there exists a finite $t^*$ for which $W(\mathcal{C}(\hat{\boldsymbol{\pi}}^{t^*}), \mathcal{D}) - \hat{\eta}^{t^*} < \varepsilon$.*

## 3.2 ACCELERATING THE ALGORITHM

Since W-RMP($\Lambda$) is a relaxation of W-MP, we can accelerate the algorithm with techniques to tighten this relaxation. We propose two families of constraints which can be added to W-RMP($\Lambda$).

**Enhanced Optimality Cuts (EOC).** A given inequality is referred to as an optimality cut for an optimization problem if the optimal solution is guaranteed to satisfy the inequality. Introducing optimality cuts as additional constraints to an optimization problem will not change the optimal solution, but it may lead to a smaller feasible set. We augment W-RMP($\Lambda$) with "Enhanced Optimality Cuts" (EOC) in addition to the Lagrangian constraints at each iteration of the algorithm.

**Proposition 1.** *For all $\hat{\boldsymbol{\pi}} \in \mathcal{P}$, let $(\hat{\boldsymbol{\Gamma}}, \hat{\boldsymbol{\lambda}}, \hat{\boldsymbol{\mu}})$ be optimal primal-dual solutions to $W(\mathcal{C}(\hat{\boldsymbol{\pi}}), \mathcal{D})$ in problems (1) and (2). Let $D_{i,i'}$ be elements of $\mathbf{D_x}$. The following are optimality cuts for W-MP:*

*Lower Bound:*
$$\eta \geq \frac{1}{B} \sum_{i=1}^{N} \min_{i' \in \{1, \cdots, N\}} \{D_{i,i'} | D_{i,i'} > 0\} \pi_i \tag{6}$$

*Triangle Inequality:*
$$\eta \geq \frac{1}{B} \sum_{i=1}^{N} \min_{\substack{i' \in \{1, \cdots, N\} \\ \mathbf{x}_{i'} \in \mathcal{C}(\hat{\boldsymbol{\pi}})}} \{D_{i,i'} | D_{i,i'} > 0\} \pi_i - W(\mathcal{C}(\hat{\boldsymbol{\pi}}), \mathcal{D}) \tag{7}$$

*Dual Inequality:*
$$\eta \geq W(\mathcal{C}(\hat{\boldsymbol{\pi}}), \mathcal{D}) + \frac{1}{B} \hat{\boldsymbol{\lambda}}^{\mathsf{T}} \boldsymbol{\pi} - \frac{1}{N} \hat{\boldsymbol{\mu}}^{\mathsf{T}} \mathbf{1} - \frac{1}{B} \sum_{i=1}^{N} \max_{i' \in \{1, \cdots, N\}} \{D_{i,i'}\} \boldsymbol{\pi}_i \tag{8}$$

Inequality (6) is derived from a lower bound on the Wasserstein distance between two probability distributions. This is a general relationship between $\eta$ and $\boldsymbol{\pi}$ and can be included as an additional constraint to W-RMP($\Lambda$) at the onset of the algorithm. Inequalities (7) and (8) bound $\eta$ and $\boldsymbol{\pi}$ with respect to a given $\hat{\boldsymbol{\pi}}$. In each iteration, when we obtain a new $\hat{\boldsymbol{\pi}}$, we can construct a Triangle and Dual Inequality, respectively, and add them to W-RMP($\Lambda$) along with the Lagrangian constraints.

From Proposition 1, an optimal solution to W-MP satisfies the inequalities (6)–(8), meaning that including them as additional constraints to W-RMP($\Lambda$) will still ensure that the optimal solution to W-MP is feasible for W-RMP($\Lambda$) for any $\Lambda$ (and therefore also optimal once $\Lambda$ is sufficiently large). That is, we preserve convergence from Corollary 1, while tightening the relaxation, which reduces the search space and consequently, the overall number of iterations of our algorithm.

**Pruning Constraints (P).** Augmenting W-RMP($\Lambda$) with constraints that are not valid optimality cuts may potentially remove the global optimal solution of W-MP from the relaxed problem. Nonetheless, such constraints may empirically improve the algorithm by aggressively removing unnecessary regions of the relaxed feasible set (Rahmaniani et al., 2017). One simple approach is to use trust region-type constraints that prune neighbourhoods of good versus bad solutions (van Ackooij et al., 2016). We introduce a simple set of constraints that remove search neighbourhoods around core sets with high Wasserstein distances and encourage searching near neighbourhoods around core sets with lower Wasserstein distances. We detail these constraints in Appendix D.1.

In Appendix D.2, we provide a full algorithm description and Python pseudo-code. Finally, note that the techniques proposed here are not exhaustive and more can be used to improve the algorithm.

## 4 COMPARISON WITH RELATED LITERATURE

In this section, we compare our optimization problem and active learning strategy with the related literature in active learning, minimizing Wasserstein distances, self-supervised learning, and GBD.

**Active Learning.** Our approach is closely related to Greedy $k$-centers (Sener & Savarese, 2017), which develops a core set bound similar to Theorem 1 but using the optimal solution to a $k$-center facility location problem rather than the Wasserstein distance (Wolf, 2011). Their bound is probabilistic whereas ours is deterministic. Furthermore since facility location is a large MILP, Sener & Savarese (2017) use a greedy approximation to select points. This sub-optimality can incur significant opportunity cost in settings where every point must be carefully selected so as to not waste a limited budget. Here, GBD solves our MILP with an optimality guarantee if given sufficient runtime.

Another related method is Wasserstein Adversarial Active learing (WAAL) (Shui et al., 2020), which also proposes a probabilistic Wasserstein distance core set bound. Their bound is proved with techniques from domain adaptation and requires a Probabilistic-Lipschitz assumption on the data distribution itself, whereas our bound uses only linear duality and a weaker assumption: Lipschitz continuity in the loss function. To select points, Shui et al. (2020) use a semi-supervised adversarial regularizer to estimate the Wasserstein distance and a greedy heuristic to select points that minimize the regularizer. Our algorithm does not need to learn a custom feature space and can instead be used with any features. Furthermore, we directly optimize the Wasserstein distance and provide a theoretical optimality gap. Numerical results show that optimization outperforms greedy estimation in the low-budget regime and yields a small improvement even in high-budget settings.

**Minimizing Wasserstein Distance.** Problem (3) has been previously studied in active learning (Ducoffe, 2018; Shui et al., 2020) and in other applications such as scenario reduction for stochastic optimization (Heitsch & Römisch, 2003; Rujeerapaiboon et al., 2018). The related literature concedes that MILP models for this problem have difficulty scaling to large data sets, necessitating heuristics. For instance in scenario reduction, Heitsch & Römisch (2003) and Rujeerapaiboon et al. (2018) equate problem (3) to the $k$-Medoids Cluster Center optimization problem. They employ the $k$-medoids heuristic and an approximation algorithm, respectively, on small data sets containing approximately $1,000$ samples. In contrast, our GBD algorithm provides an optimality guarantee and as we show in numerical experiments, scales gracefully on data sets with $75,000$ samples. We review $k$-medoids and scenario reduction in Appendix E.2.

**Self-supervised Learning.** Active learning selection strategies improve with useful features $\mathbf{x} \in \mathcal{X}$. Recently, self-supervised learning has shown tremendous promise in extracting representations

for semi-supervised tasks. For example, a model pre-trained with SimCLRv2 on ImageNet (Deng et al., 2009) can be finetuned with labels from $1\%$ of the data to achieve impressive classification accuracy (Chen et al., 2020a;b;c). In contrast, the standard practice in classical active learning is to train a classifier for feature extraction with previously labeled data (Sener & Savarese, 2017; Sinha et al., 2019; Shui et al., 2020), with the previous papers requiring labeling up to $20\%$ of the data to achieve comparable performance to self-supervised pre-training with supervised finetuning. Our numerical experiments verify that self-supervised pre-training can accelerate active learning to competitive accuracy with orders of magnitude less labeled data.

Chen et al. (2020c) use random selection after self-supervised learning. With enough points, current active learning strategies achieve similar bottleneck performances (Siméoni et al., 2019). However, the heuristic nature of these strategies leave opportunities in the low budget regime. Numerical results show that optimization in this regime specifically yields large gains over heuristic baselines.

**Generalized Benders Decomposition (GBD).** This framework is commonly used in non-convex and integer constrained optimization (Geoffrion, 1972), since it instead solves sub-problems while often providing an optimality guarantee. However, the vanilla GBD algorithm is typically slow to converge for large problems. As a result, the integer programming literature proposes improvement techniques such as multiple different types of constraints to tighten the relaxations and heuristics to direct the search (Rahmaniani et al., 2017). These improvements are often customized for the specific optimization problem being solved. In our case, we propose enhanced optimality cuts in Section 3.2 that leverage properties of the Wasserstein distance. Further, our pruning constraints are simple techniques that can be used in GBD for arbitrary binary MILPs.

## 5 EXPERIMENTS

In this section, we first evaluate our algorithm on solving problem (4) by comparing against the standard heuristic from the scenario reduction literature (Heitsch & Römisch, 2003). We then evaluate our algorithm on the following two active learning tasks: (i) low budget image classification where we select images based on features obtained via self-supervised learning and (ii) domain adaptation where we select images in an unlabeled target pool using features obtained via unsupervised domain adaptation over a source data pool. For both tasks, we outperform or remain competitive with the best performing baselines on all data sets and budget ranges.

In ablations, we first observe that our approach is effective with different embedding methods (e.g., self-supervised, domain adversarial, and classical active learning where features are obtained from previous rounds); in each case, we yield improvements over baselines. Second, the acceleration techniques proposed in Section 3.2 help optimize problem (4) faster and often improve active learning. Finally, from a wall-clock analysis, running GBD for more iterations yields better selections, but even early stopping after 20 minutes is sufficient for high-quality solutions.

In this section, we only summarize the main results. We leave detailed analyses for all experiments in Appendix E. Further in Appendix E.7, we include additional experiments including ablations and qualitative evaluation of the selections and $t$-SNE visualizations of the feature space.

### 5.1 EVALUATION PROTOCOL AND BASELINES

We evaluate active learning in classification on three data sets: STL-10 (Coates et al., 2011), CIFAR-10 (Krizhevsky, 2009), and SVHN (Netzer et al., 2011). Each data set contains $5,000$, $50,000$, and $73,257$ images, respectively, showcasing when the number of potential core sets ranges from small to large. We initialize with no labeled points and first pre-train our model (i.e., feature encoder + head) using the self-supervised method SimCLR (Chen et al., 2020a;b). We then employ active learning over sequential rounds. In each round, we use feature vectors obtained from the encoder output to select and label $B$ new images and then train the classifier with supervised learning.

Our domain adaptation experiments use the Office-31 data set (Saenko et al., 2010), which contains $4,652$ images from three domains, Amazon (A), Web (W), and DSLR (D). We use active learning to label points from an initially unlabeled target set to maximize accuracy over the target domain. We follow the same sequential procedure as above except with a different feature encoding step, $f$-DAL, which is an unsupervised domain adversarial learning method (Acuna et al., 2021).

Table 1: Head-to-head comparison of our solver using Enhanced Optimality Cuts (EOC) versus $k$-medoids on the objective function value of problem (4) at different budgets. Lower values are better. For each data set, the best solution at each budget is bolded and underlined. All entries show means over five runs. See Appendix E.2 for standard deviations.

|  | $B$ | 10 | 20 | 40 | 60 | 80 | 100 | 120 | 140 | 160 | 180 |
|---|---|---|---|---|---|---|---|---|---|---|---|
| STL-10 | $k$-medoids | 0.290 | 0.132 | **0.109** | **0.110** | **0.100** | 0.097 | **0.092** | 0.089 | 0.093 | 0.087 |
|  | Wass. + EOC | **0.281** | **0.134** | 0.122 | 0.116 | 0.101 | **0.096** | **0.092** | **0.087** | **0.090** | **0.085** |
| CIFAR-10 | $k$-medoids | **0.258** | **0.142** | **0.118** | 0.120 | 0.109 | 0.104 | 0.095 | 0.089 | 0.099 | 0.096 |
|  | Wass. + EOC | 0.322 | 0.176 | 0.125 | **0.092** | **0.077** | **0.097** | **0.064** | **0.059** | **0.072** | **0.072** |
| SVHN | $k$-medoids | **0.152** | 0.172 | 0.164 | 0.175 | 0.178 | 0.179 | 0.178 | 0.173 | 0.173 | 0.170 |
|  | Wass. + EOC | 0.153 | **0.093** | **0.093** | **0.100** | **0.098** | **0.091** | **0.085** | **0.080** | **0.087** | **0.089** |

We use a ResNet-18 (He et al., 2016) feature encoder for STL-10, ResNet-34 for CIFAR-10 and SVHN, and Resnet-50 for Office-31. Our head is a two-layer MLP. The downstream classifier is the pre-trained encoder (whose weights are frozen) with a new head in each round. Supervised learning is performed with cross entropy loss. We compute Wasserstein distances using the Python Optimal Transport library (Flamary et al., 2021) and solve W-RMP($\Lambda$) with the COIN-OR/CBC Solver (Forrest & Lougee-Heimer, 2005). We run our algorithm up to a time limit of 3 hours (and increase to 6 hours for experiments on SVHN). Finally, we implement two variants of the GBD algorithm without the pruning constraints (Wass. + EOC) and with them (Wass. + EOC + P). See Appendix E for details on training and optimizer settings.

We compare against the following baselines: (i) Random Selection; (ii) Least Confidence (Settles, 2009; Shen et al., 2017; Gal et al., 2017); (iii) Maximum Entropy (Settles, 2012); (iv) $k$-Medoids Cluster Centers (Heitsch & Römisch, 2003); (v) Greedy $k$-centers (Sener & Savarese, 2017); (vi) Wasserstein Adversarial Active Learning (WAAL) (Shui et al., 2020). The second and third baselines are uncertainty-based approaches using the trained classifier in each round, the fourth and fifth are representation-based approaches that leverage unsupervised pre-trained features, and the last is a greedy strategy that approximates the Wasserstein distance with an adversarial regularizer in training. All baselines use the same self-supervised pre-training and supervised learning steps as us.

## 5.2 MAIN RESULTS

**Objective Function Value (see Appendix E.2 for more results).** $k$-Medoids is a well-known heuristic for problem (4) (Heitsch & Römisch, 2003; Rujeerapaiboon et al., 2018). Table 1 compares objective function values obtained by our algorithm versus $k$-medoids for STL-10, CIFAR-10, and SVHN. We especially outperform $k$-medoids for $B \geq 120$ where the selection problem is large and harder to solve. This gap also widens for larger data sets; for instance on SVHN, we almost always achieve around $2\times$ improvement in objective function value. Table 2 further compares $k$-medoids versus Wass. + Enhanced Optimality Cuts (EOC) on CIFAR-10 where we force early termination. Even after reducing our algorithm's runtime to three minutes, we still achieve high-quality solutions to the optimization problem.

Table 2: Wass. + EOC with limited runtimes versus $k$-medoids on problem (4) for CIFAR-10. The best solution for each budget is bolded and underlined. See Appendix E.2 for full results and Appendix E.6 for details on $k$-medoids runtime.

| $B$ | $k$-medoids $\approx$ 3 min. | Wass. + EOC 3 min. | 3 hr. |
|---|---|---|---|
| 120 | 0.095 | 0.082 | **0.064** |
| 140 | 0.089 | 0.073 | **0.059** |
| 160 | 0.099 | 0.084 | **0.072** |
| 180 | 0.096 | 0.075 | **0.072** |

**Image Classification (Appendix E.3).** Figure 2 summarizes our methods versus baselines on each data set. For each data set and number of points, our approaches are either competitive with or outperform the best baseline. Table 3 highlights the extremely low budget regime ($B \leq 40$) where every point must be carefully selected. For instance, if we can only label $B = 10$ points, poorly selecting even one point wastes $10\%$ of the budget. Here, our best methods typically outperform the baselines on all three data sets often by large margins (e.g., $\geq 4.9\%$ on STL-10 at $B = 40$, $\geq 9.1\%$ on CIFAR-10 at $B = 20$, and $\geq 5.2\%$ on SVHN at $B = 40$).

There is no dominant baseline over all budget ranges and data sets: $k$-medoids is effective for $B \leq 40$ or smaller data sets (STL-10) but worse with bigger budgets and data sets (CIFAR-10, SVHN), whereas Random and Greedy $k$-centers improve in the latter. Furthermore while SimCLR-pretraining improves all baselines compared to classical active learning papers (Sener & Savarese,

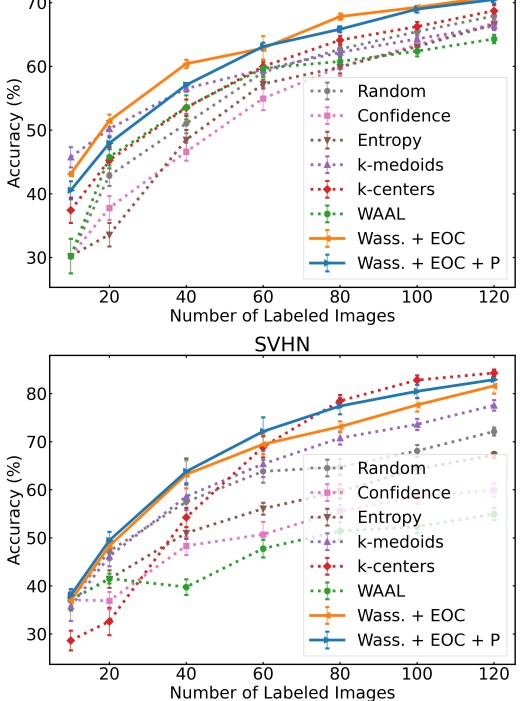
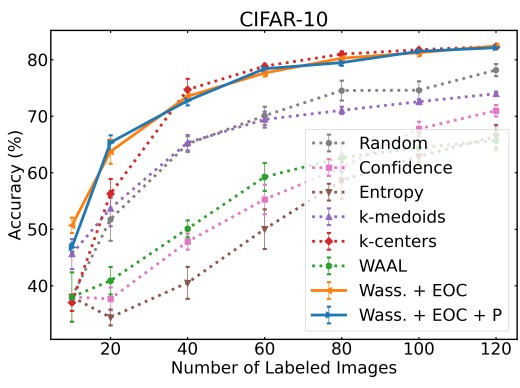

Figure 2: Low budget active learning on CIFAR-10, SVHN and STL-10 using SimCLR pre-trained features. See Table 3 for detailed results for $B \leq 40$. See Appendix E.3 for full results up to $B \leq 180$. The solid lines are our models and the dashed lines are baselines. All plots show mean $\pm$ standard error over five runs. Our best models outperform baselines for most budgets.

Table 3: Mean $\pm$ standard deviation of accuracy results in the extremely low budget regime.

| Data set | $B$ | Random | Confidence | Entropy | $k$-medoids | $k$-centers | WAAL | Wass. + EOC | Wass. + EOC + P |
|---|---|---|---|---|---|---|---|---|---|
| STL-10 | 10 | $30.2 \pm 6.1$ | $30.2 \pm 6.1$ | $30.2 \pm 6.1$ | $\mathbf{45.7 \pm 3.5}$ | $37.4 \pm 4.4$ | $30.2 \pm 6.1$ | $\underline{43.1 \pm 0.7}$ | $41.4 \pm 3.2$ |
|  | 20 | $42.8 \pm 3.6$ | $37.8 \pm 4.2$ | $33.6 \pm 4.1$ | $\underline{50.2 \pm 2.4}$ | $45.3 \pm 4.6$ | $45.8 \pm 4.0$ | $\mathbf{51.5 \pm 2.1}$ | $49.5 \pm 3.6$ |
|  | 40 | $51.0 \pm 3.4$ | $46.6 \pm 3.1$ | $48.5 \pm 3.4$ | $56.5 \pm 1.2$ | $53.5 \pm 2.9$ | $53.6 \pm 4.2$ | $\mathbf{60.4 \pm 1.5}$ | $\underline{58.7 \pm 1.8}$ |
| CIFAR-10 | 10 | $38.0 \pm 9.7$ | $38.0 \pm 9.7$ | $38.0 \pm 9.7$ | $45.6 \pm 5.9$ | $37.0 \pm 3.2$ | $38.0 \pm 9.7$ | $\mathbf{50.7 \pm 3.0}$ | $\underline{46.8 \pm 1.6}$ |
|  | 20 | $51.8 \pm 8.5$ | $37.7 \pm 4.5$ | $34.4 \pm 3.1$ | $53.7 \pm 7.0$ | $56.2 \pm 5.9$ | $40.9 \pm 5.5$ | $\underline{63.7 \pm 4.8}$ | $\mathbf{65.3 \pm 2.9}$ |
|  | 40 | $65.1 \pm 3.1$ | $47.8 \pm 3.1$ | $40.5 \pm 6.3$ | $65.3 \pm 3.0$ | $\mathbf{74.7 \pm 4.3}$ | $50.1 \pm 3.3$ | $\underline{73.5 \pm 1.6}$ | $72.7 \pm 1.8$ |
| SVHN | 10 | $37.1 \pm 4.9$ | $37.1 \pm 4.9$ | $37.1 \pm 4.9$ | $35.7 \pm 6.7$ | $28.6 \pm 4.6$ | $\underline{37.1 \pm 4.9}$ | $37.0 \pm 2.8$ | $\mathbf{38.0 \pm 2.9}$ |
|  | 20 | $47.2 \pm 4.2$ | $36.9 \pm 4.1$ | $41.4 \pm 4.1$ | $46.1 \pm 4.6$ | $32.6 \pm 6.4$ | $41.5 \pm 2.3$ | $\underline{48.3 \pm 3.6}$ | $\mathbf{49.4 \pm 4.0}$ |
|  | 40 | $57.5 \pm 6.3$ | $48.3 \pm 4.3$ | $51.0 \pm 2.8$ | $58.6 \pm 6.3$ | $54.2 \pm 4.4$ | $39.8 \pm 3.6$ | $\underline{63.2 \pm 6.5}$ | $\mathbf{63.8 \pm 6.1}$ |

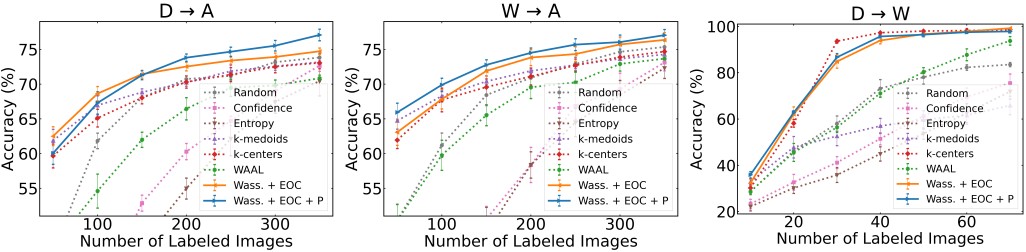

Figure 3: Active learning with domain adaptation for D→A, W→A, and D→W on Office-31 using $f$-DAL pre-trained features. The solid line is our best model and the dashed lines are baselines. All plots show mean $\pm$ standard error over three runs. See Appendix E.4 for full results.

2017; Shui et al., 2020), representation methods (Greedy $k$-centers, $k$-medoids) generally outperform uncertainty methods (WAAL, Least Confidence, Maximum Entropy) in this setup because classification is now easier, which means that it is more useful to obtain representative samples of each class than uncertain examples. By optimizing for a diverse core set selection, we outperform with the best baseline for nearly all budgets and data sets.

**Domain Adaptation (Appendix E.4).** Figure 3 previews accuracy on Office-31 for DSLR to Amazon, Web to Amazon, and DSLR to Web; we leave the other source-target combinations in Appendix E.4. As before, our approach either outperforms or is competitive with the best baseline for all budgets and source-target pairs, the baselines are non-dominating over each other, and representation methods such as ours benefit from strong features and model pre-training. Thus, our

Table 4: Mean ± standard deviation of accuracy results in the classic active learning setup of Shui et al. (2020). The best method is bolded and underlined.

| | $B$ | 1000 | 2000 | 3000 | 4000 | 5000 | 6000 |
|---|---|---|---|---|---|---|---|
| CIFAR-10 | WAAL | $57.0 \pm 1.2$ | $65.9 \pm 0.4$ | $\mathbf{\underline{73.3 \pm 0.6}}$ | $76.7 \pm 0.8$ | $77.4 \pm 0.7$ | $\mathbf{\underline{81.0 \pm 0.2}}$ |
| | Wass. + EOC + P | $\mathbf{\underline{58.1 \pm 0.1}}$ | $\mathbf{\underline{68.9 \pm 0.7}}$ | $73.2 \pm 0.2$ | $\mathbf{\underline{77.3 \pm 0.3}}$ | $\mathbf{\underline{79.1 \pm 0.7}}$ | $81.0 \pm 0.4$ |
| | $B$ | 500 | 1000 | 1500 | 2000 | 2500 | 3000 |
| SVHN | WAAL | $\mathbf{\underline{69.6 \pm 0.9}}$ | $76.9 \pm 0.8$ | $84.2 \pm 0.7$ | $86.2 \pm 1.7$ | $87.5 \pm 1.0$ | $88.9 \pm 0.5$ |
| | Wass. + EOC + P | $69.3 \pm 1.1$ | $\mathbf{\underline{79.4 \pm 1.7}}$ | $\mathbf{\underline{84.7 \pm 0.6}}$ | $\mathbf{\underline{86.3 \pm 0.6}}$ | $\mathbf{\underline{88.3 \pm 0.8}}$ | $\mathbf{\underline{89.9 \pm 0.5}}$ |

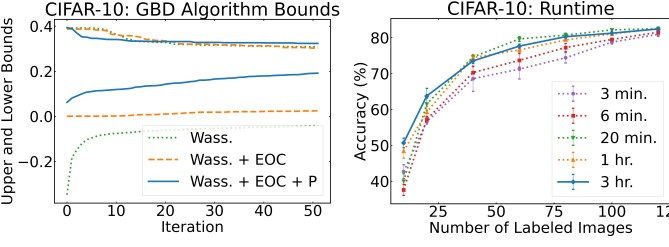

Figure 4: (Left) Upper and lower bounds in the first 50 iterations at $B = 10$. (Right) Increasing the wall-clock runtime of the GBD algorithm. All plots show mean ± standard error over three runs.

framework can be plugged into any classification task as long as there is a good feature space for optimization; for example, here we use unsupervised domain adaptation (Acuna et al., 2021).

**Representations from classical active learning (Appendix E.5).** Table 4 ablates the effect of using features from unsupervised pre-training by presenting accuracy on CIFAR-10 and SVHN under classical active learning. We implement the exact setup of and compare against WAAL (Shui et al., 2020), which was specifically designed for the classical setting. Although the optimization problem is harder than in the low-budget setting due to having less useful features and solving for larger $B$, our approach is still competitive and can outperform WAAL sometimes by up to $3\%$. This suggests that our algorithm is also effective in classical active learning.

**Acceleration Techniques.** Figure 4 (left) plots the upper and lower bounds for the first 50 iterations of the vanilla algorithm Wass., Wass. + EOC, and Wass. + EOC + P (i.e. including Pruning constraints) for CIFAR-10 at $B = 10$. The lower bounds for Wass. are negative, meaning the vanilla algorithm requires a large number of iterations to improve the bounds. Wass. + EOC and Wass. + EOC + P both improve the lower bounds. At the same time, Wass. + EOC + P yields better upper bounds early in the algorithm, but is restricted to a local minimum whereas Wass. + EOC eventually improves over the heuristic. Thus, Wass. requires many iterations to converge, necessitating our cuts and pruning techniques.

**Wall-clock Analysis (Appendix E.6).** Our algorithm is effective even without convergence. Figure 4 (right) plots accuracy on CIFAR-10 for Wass. + EOC where we terminate at different runtimes and use the incumbent solution. Longer runtimes typically yield better results, especially for $B \leq 40$ (i.e., a sub-optimal selection here wastes $\geq 2.5\%$ of the budget). However at higher budgets, setting the runtime as low as 20 minutes is sufficient to obtain high-quality selections, since the marginal value of each new labeled point decreases as the labeled set increases. Note that the baselines typically require between several seconds to minutes to make selections, meaning our algorithm is on average slower to generate selections. Nonetheless, the time cost of selection is still less than model training time, and for some applications, less than human labeling time (Rajchl et al., 2016), meaning a slightly longer selection time is generally acceptable if we can yield better selections.

## 6 DISCUSSION

When the budget for labeling images is limited, every point must be carefully selected. In this work, we propose an integer optimization problem for selecting a core set of data to label. We prove that our problem upper bounds the expected risk, and we develop a GBD algorithm for solving our problem. Under relevant features, formulating a MILP ensures that we may find globally optimal core set selections, in contrast to most selection strategies that serve as efficient but ad-hoc heuristics. Our numerical results show that our algorithm is an effective minimizer of the Wasserstein distance compared to heuristics. Finally, we show over four image data sets that our optimization is competitive with current baselines in active learning, especially when the labeling budget is low.

ACKNOWLEDGMENTS

We would like to thank David Acuna, James Lucas and the anonymous reviewers for feedback on earlier versions of this paper, and Andre Cire and Jean-François Puget for early discussions on Benders decomposition.

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

## APPENDIX

## A  PROOFS OF STATEMENTS

*Proof of Theorem 1.* Let $\mathbf{D}_y = [|\Omega(\mathbf{x}_i) - \Omega(\mathbf{x}_i')|]_{i=1,i'=1}^{N,N}$ be a distance matrix between the labels of points in $\mathcal{D}$. Let $\mathcal{L}_K$ be the set of $K$-Lipschitz functions. Using the Kantorovich-Rubenstein Duality (Villani, 2008), we bound the core set loss by a Wasserstein distance

$$
\frac{1}{N}\sum_{i=1}^{N}\ell(\mathbf{x}_i,\Omega(\mathbf{x}_i);\mathbf{w}) - \frac{1}{B}\sum_{j=1}^{B}\ell(\mathbf{x}_j,\Omega(\mathbf{x}_j);\mathbf{w})
$$

$$
\leq \sup_{\tilde{\ell}\in\mathcal{L}_K}\left\{\frac{1}{N}\sum_{i=1}^{N}\tilde{\ell}(\mathbf{x}_i,\Omega(\mathbf{x}_i);\mathbf{w}) - \frac{1}{B}\sum_{i'=1}^{N}\pi_{i'}\tilde{\ell}(\mathbf{x}_{i'},\Omega(\mathbf{x}_{i'});\mathbf{w})\right\}
$$

$$
= \begin{cases} \max\limits_{\boldsymbol{\lambda},\boldsymbol{\mu}} & \frac{1}{N}\boldsymbol{\mu}^\mathsf{T}\mathbf{1} - \frac{1}{B}\boldsymbol{\lambda}^\mathsf{T}\boldsymbol{\pi} \\ \text{s.\,t.} & \boldsymbol{\lambda}^\mathsf{T}\otimes\mathbf{1} + \boldsymbol{\mu}\otimes\mathbf{1}^\mathsf{T} \leq K(\mathbf{D_x}+\mathbf{D}_y) \end{cases}
$$

$$
= \begin{cases} \min\limits_{\boldsymbol{\Gamma}\geq\mathbf{0}} & K\langle(\mathbf{D_x}+\varepsilon\mathbf{D}_y),\boldsymbol{\Gamma}\rangle \\ \text{s.\,t.} & \boldsymbol{\Gamma}\mathbf{1} = \frac{1}{N}\mathbf{1} \\ & \boldsymbol{\Gamma}^\mathsf{T}\mathbf{1} = \frac{1}{B}\boldsymbol{\pi}. \end{cases}
$$

The first line introduces probability masses and upper bounds the objective with a supremum over all $K$-Lipschitz functions $\tilde{\ell}\in\mathcal{L}_K$. The second line replaces $\tilde{\ell}$ with vectors $\boldsymbol{\mu}$ and $\boldsymbol{\lambda}$ that satisfy the Lipschitz property. The final line takes the dual of the linear program.

Next note that $|\Omega(\mathbf{x}_i) - \Omega(\mathbf{x}_{i'})| \leq C$, meaning that for any feasible $\boldsymbol{\Gamma}$ to the Wasserstein problem, we have the following inequality:

$$
\langle\mathbf{D}_y,\boldsymbol{\Gamma}\rangle = \sum_{i=1}^{N}\sum_{i'=1}^{N}|\Omega(\mathbf{x}_i) - \Omega(\mathbf{x}_{i'})|\Gamma_{i,i'}
$$

$$
\leq C\sum_{i=1}^{N}\sum_{i'=1}^{N}\Gamma_{i,i'} = C.
$$

The last equality follows from the constraints in the Wasserstein distance primal program implying $\mathbf{1}^\mathsf{T}\boldsymbol{\Gamma}\mathbf{1} = 1$. We now rewrite the objective of the linear program to

$$
K\langle\mathbf{D_x}+\varepsilon\mathbf{D}_y,\boldsymbol{\Gamma}\rangle = K\left(\langle\mathbf{D_x},\boldsymbol{\Gamma}\rangle + \varepsilon\langle\mathbf{D}_y,\boldsymbol{\Gamma}\rangle\right)
$$

$$
\leq K\langle\mathbf{D_x},\boldsymbol{\Gamma}\rangle + KC\varepsilon.
$$

Since the second term does not depend on $\boldsymbol{\Gamma}$, we remove it from the optimization objective and complete the proof. $\square$

*Proof of Corollary 1.* The proof follows from Lemma 2 (see Appendix C), which states that if the objectives and constraints of the optimization problem are (i) separable in the decision variables and (ii) linear, then the algorithm will converge. Problem (4) satisfies the requirements of this lemma, meaning that our algorithm terminates in finite iterations. $\square$

*Proof of Proposition 1.* Let $(\eta^*,\boldsymbol{\pi}^*)$ be the optimal solution to W-MP and let $\boldsymbol{\Gamma}^*$ be the optimal primal solution to $W(\mathcal{C}(\boldsymbol{\pi}^*),\mathcal{D})$. To prove that each inequality is an optimality cut, we show that

they are met by $(\eta^*, \boldsymbol{\pi}^*)$. We first prove the Lower Bound:

$$
\eta^* \geq \langle \mathbf{D_x}, \boldsymbol{\Gamma}^* \rangle + \boldsymbol{\lambda}^{*\mathsf{T}} \left( \frac{1}{B} \boldsymbol{\pi}^* - \boldsymbol{\Gamma}^{*\mathsf{T}} \mathbf{1} \right)
$$

$$
= \sum_{i=1}^{N} \sum_{i'=1}^{N} D_{i,i'} \Gamma_{i,i'}^*
$$

$$
\geq \sum_{i=1}^{N} \min_{i' \in \{1,\ldots,N\}} \{ D_{i,i'} | D_{i,i'} > 0 \} \sum_{i'=1}^{N} \Gamma_{i,i'}^*
$$

$$
= \sum_{i=1}^{N} \min_{i' \in \{1,\ldots,N\}} \{ D_{i,i'} | D_{i,i'} > 0 \} \frac{1}{B} \pi_i^*
$$

The second line follows from expanding the trace and noting $\boldsymbol{\Gamma}^{*\mathsf{T}} \mathbf{1} = \boldsymbol{\pi}^*/B$. The inequality follows from taking the minimum over all $D_{i,i'}$ and the last line follows from $\boldsymbol{\Gamma}^{*\mathsf{T}} \mathbf{1} = \boldsymbol{\pi}^*/B$.

To prove the Triangle Inequality, note that the Wasserstein distance is a distance metric and then satisfies a triangle inequality by definition. That is, for any $\hat{\boldsymbol{\pi}} \in \mathcal{P}$,

$$
W(\mathcal{C}(\boldsymbol{\pi}^*), \mathcal{D}) + W(\mathcal{C}(\hat{\boldsymbol{\pi}}), \mathcal{D}) \geq W(\mathcal{C}(\boldsymbol{\pi}^*), \mathcal{C}(\hat{\boldsymbol{\pi}})). \tag{9}
$$

The right-hand-side of the inequality is a Wasserstein distance which can be lower bounded similar to before:

$$
W(\mathcal{C}(\boldsymbol{\pi}^*), \mathcal{C}(\hat{\boldsymbol{\pi}})) =
\begin{cases}
\min_{\boldsymbol{\Gamma} \geq \mathbf{0}} & \sum_{i=1}^{N} \sum_{\substack{i'=1 \\ \mathbf{x}_{i'} \in \mathcal{C}(\hat{\boldsymbol{\pi}})}}^{N} D_{i,i'} \Gamma_{i,i'} \\
\text{s.t.} & \boldsymbol{\Gamma} \mathbf{1} = \frac{1}{B} \boldsymbol{\pi}^* \\
& \boldsymbol{\Gamma}^{\mathsf{T}} \mathbf{1} = \frac{1}{B} \hat{\boldsymbol{\pi}}
\end{cases}
$$

$$
\geq
\begin{cases}
\min_{\boldsymbol{\Gamma} \geq \mathbf{0}} & \sum_{i=1}^{N} \min_{\substack{i' \in \{1,\ldots,N\} \\ \mathbf{x}_{i'} \in \mathcal{C}(\hat{\boldsymbol{\pi}})}} \{ D_{i,i'} | D_{i,i'} > 0 \} \sum_{i'=1}^{N} \Gamma_{i,i'} \\
\text{s.t.} & \boldsymbol{\Gamma} \mathbf{1} = \frac{1}{B} \boldsymbol{\pi}^* \\
& \boldsymbol{\Gamma}^{\mathsf{T}} \mathbf{1} = \frac{1}{B} \hat{\boldsymbol{\pi}}
\end{cases}
$$

$$
= \sum_{i=1}^{N} \min_{\substack{i' \in \{1,\ldots,N\} \\ \mathbf{x}_{i'} \in \mathcal{C}(\hat{\boldsymbol{\pi}})}} \{ D_{i,i'} | D_{i,i'} > 0 \} \frac{1}{B} \pi_i^*.
$$

The key difference in this sequence from the previous is that we only consider indices $i'$ where $\mathbf{x}_{i'} \in \mathcal{C}(\hat{\boldsymbol{\pi}})$. Finally, recall that $\eta^* \geq W(\mathcal{C}(\boldsymbol{\pi}^*), \mathcal{D})$. Substituting these two inequalities and rearranging (9) completes the proof.

Finally, to prove the Dual Inequality, note that

$$
\eta^* \geq W(\mathcal{C}(\hat{\boldsymbol{\pi}}), \mathcal{D}) - W(\mathcal{C}(\hat{\boldsymbol{\pi}}), \mathcal{D}) + W(\mathcal{C}(\boldsymbol{\pi}^*), \mathcal{D}) \tag{10}
$$

We lower bound the second term by obtaining an upper bound on the Wasserstein distance:

$$
W(\mathcal{C}(\hat{\boldsymbol{\pi}}), \mathcal{D}) = \sum_{i=1}^{N} \sum_{i'=1}^{N} D_{i,i'} \hat{\Gamma}_{i,i'}
$$

$$
\leq \sum_{i=1}^{N} \max_{i' \in \{1,\ldots,N\}} \{ D_{i,i'} \} \sum_{i'=1}^{N} \hat{\Gamma}_{i,i'}
$$

$$
= \sum_{i=1}^{N} \max_{i' \in \{1,\ldots,N\}} \{ D_{i,i'} \} \frac{1}{B} \hat{\pi}_i
$$

where the upper bound now follows from taking a maximum. Subtracting from both sides lower bounds the second term in (10). Finally, we lower bound $(\mathcal{C}(\boldsymbol{\pi}^*), \mathcal{D})$ by considering the Wasserstein distance dual program

$$W(\mathcal{C}(\boldsymbol{\pi}^*), \mathcal{D}) = \begin{cases} \max_{\boldsymbol{\lambda}, \boldsymbol{\mu}} & \frac{1}{N} \boldsymbol{\mu}^\top \mathbf{1} - \frac{1}{B} \boldsymbol{\lambda}^\top \boldsymbol{\pi}^* \\ \text{s.t.} & \boldsymbol{\lambda}^\top \otimes \mathbf{1} + \boldsymbol{\mu} \otimes \mathbf{1}^\top \leq \mathbf{D_x} \end{cases}$$

$$\geq \frac{1}{N} \hat{\boldsymbol{\mu}}^\top \mathbf{1} - \frac{1}{B} \hat{\boldsymbol{\lambda}}^\top \boldsymbol{\pi}^*$$

Note that the optimal dual variables $(\hat{\boldsymbol{\lambda}}, \hat{\boldsymbol{\mu}})$ for $W(\mathcal{C}(\hat{\boldsymbol{\pi}}), \mathcal{D})$ are feasible solutions for the Wasserstein dual problem $W(\mathcal{C}(\boldsymbol{\pi}^*), \mathcal{D})$ and thus, also a lower bound on the Wasserstein distance. Substituting the corresponding objective value into (10) completes the proof. $\qquad \square$

## B  ON THE LIPSCHITZ CONTINUITY ASSUMPTION

Our main result in Theorem 1 states that the Wasserstein distance upper bounds the core set loss if the training loss function $\ell(\mathbf{x}, y; \mathbf{w})$ is $K$-Lipschitz continuous over the data set space $\mathcal{X} \times \mathcal{Y}$ for fixed model parameters $\mathbf{w}$. Lipschitz continuity assumptions are commonplace in the development of active learning theory (Sener & Savarese, 2017; Shui et al., 2020). We show below that our assumption is relatively mild and can sometimes be satisfied in practice.

Our assumption builds from Sener & Savarese (2017) who require their loss function $\ell(\mathbf{x}, y; \mathbf{w})$ to be Lipschitz continuous over the input features $\mathcal{X}$ only. They show that convolutional neural networks trained using Mean-Squared Error (MSE) loss meet their assumption. Although we require the loss function to be Lipschitz in both the input and output, we can extend the result of Sener & Savarese (2017) to show that convolutional neural networks are Lipschitz in the input-output space as well. We first state the Lemma of Sener & Savarese (2017) before proving our corollary.

**Lemma 1** (Sener & Savarese (2017)). *Consider the $C$-class classification problem over feature $\mathcal{X}$ and one-hot-encoded label $\hat{\mathcal{Y}} := \{\mathbf{e}_1, \ldots, \mathbf{e}_C\}$ spaces. Let $F : \mathcal{X} \to \hat{\mathcal{Y}}$ be a convolutional neural network with $n_c$ convolutional-maxpool-ReLU layers and $n_f$ fully connected layers. Then, there exists a constant $\alpha > 0$ such that the MSE loss function $\ell(\mathbf{x}, \mathbf{y}; \mathbf{w}) := \|F(\mathbf{x}; \mathbf{w}) - \mathbf{y}\|_2$ is $(\sqrt{C-1}\alpha^{n_c+n_f}/C)$-Lipschitz continuous in $\mathbf{x}$ for fixed $\mathbf{y}$ and $\mathbf{w}$.*

While the lemma states that $\|F(\mathbf{x}; \mathbf{w}) - \mathbf{y}\|_2$ is Lipschitz continuous, the proof further implies that by composition, the neural network $F(\mathbf{x}; \mathbf{w})$ is also Lipschitz continuous with the same constant.

**Corollary 2.** *Consider the metric space $(\mathcal{X} \times \mathcal{Y}, \|\mathbf{x}\| + \varepsilon|y|)$ using features $\mathcal{X}$ and labels $\mathcal{Y} = \{1, \ldots, C\}$. Let $\mathbf{e}(y)$ denote the one-hot encoding of $y$. The MSE loss function $\ell(\mathbf{x}, y; \mathbf{w}) := \|F(\mathbf{x}; \mathbf{w}) - \mathbf{e}(y)\|_2^2$ is Lipschitz continuous in $(\mathbf{x}, y)$ for fixed $\mathbf{w}$.*

*Proof.* The proof is a straightforward extension of Lemma 1. Consider two pairs $(\mathbf{x}, \mathbf{e}(y))$ and $(\mathbf{x}', \mathbf{e}(y'))$. We bound with the following steps:

$$\begin{aligned}
|\ell(\mathbf{x}, y; \mathbf{w}) - \ell(\mathbf{x}', y'; \mathbf{w})| &= \left| \|\mathbf{e}(y) - F(\mathbf{x}; \mathbf{w})\|_2 - \|\mathbf{e}(y') - F(\mathbf{x}'; \mathbf{w})\|_2 \right| \\
&\leq \|\mathbf{e}(y) - \mathbf{e}(y') + F(\mathbf{x}'; \mathbf{w}) - F(\mathbf{x}; \mathbf{w})\|_2 \\
&\leq \|\mathbf{e}(y) - \mathbf{e}(y')\|_2 + \|F(\mathbf{x}'; \mathbf{w}) - F(\mathbf{x}; \mathbf{w})\|_2 \\
&\leq |y - y'| + \frac{\sqrt{C-1}}{C} \alpha^{(n_c+n_f)} \|\mathbf{x} - \mathbf{x}'\|_2 \\
&\leq \max\left( \frac{1}{\varepsilon}, \frac{\sqrt{C-1}}{C} \alpha^{(n_c+n_f)} \right) (\varepsilon|y - y'| + \|\mathbf{x} - \mathbf{x}'\|_2)
\end{aligned}$$

The first two inequalities follow from applying the Reverse Triangle Inequality and then the Triangle Inequality. The third inequality follows from Lemma 1 and $|y - y'| \geq \|\mathbf{e}(y) - \mathbf{e}(y')\|_2$. The final inequality follows from taking the maximum of the coefficients. $\qquad \square$

Our result holds for MSE loss, specifically. In practice, we use the Cross-Entropy loss, which does not satisfy this Lipschitz continuity assumption. Nonetheless, the empirical effectiveness of our strategy highlights its usefulness.

## C  BACKGROUND ON GENERALIZED BENDERS DECOMPOSITION

Generalized Benders Decomposition (GBD) is an iterative algorithm for solving non-convex constrained optimization problems (Geoffrion, 1972). We summarize the general steps of this algorithm below and refer the reader to Rahmaniani et al. (2017) for a recent survey on this methodology.

Let $\mathbf{x} \in \mathcal{X} \subseteq \mathbb{R}^{d_1}$ and $\mathbf{y} \in \mathcal{Y} \subseteq \mathbb{R}^{d_2}$ be variables and consider the problem

$$\min_{\mathbf{x} \in \mathcal{X}, \mathbf{y} \in \mathcal{Y}} f(\mathbf{x}, \mathbf{y}) \qquad \text{s.\,t.} \ \ g(\mathbf{x}, \mathbf{y}) = \mathbf{0}, \tag{11}$$

where $f(\mathbf{x}, \mathbf{y}) : \mathbb{R}^{d_1} \times \mathbb{R}^{d_2} \to \mathbb{R}$ and $g(\mathbf{x}, \mathbf{y}) : \mathbb{R}^{d_1} \times \mathbb{R}^{d_2} \to \mathbb{R}^m$ are convex functions but $\mathcal{X}$ and $\mathcal{Y}$ may be non-convex sets. This problem is equivalent to the following bi-level Master Problem (MP):

$$\min_{\eta, \mathbf{y} \in \mathcal{Y}} \eta \qquad \text{s.\,t.} \ \ \eta \geq \inf_{\mathbf{x} \in \mathcal{X}} \left\{ f(\mathbf{x}, \mathbf{y}) + \boldsymbol{\lambda}^\mathsf{T} g(\mathbf{x}, \mathbf{y}) \right\}, \ \forall \boldsymbol{\lambda} \in \mathbb{R}^m$$

where $\eta$ is a slack variable which when minimized, is equal to the optimal value of problem (11). This MP can also be re-written with a single constraint by replacing the right-hand-side of the inequality with the Lagrangian $L(\mathbf{y}) := \sup_{\boldsymbol{\lambda} \in \mathbb{R}^m} \inf_{\mathbf{x} \in \mathcal{X}} \{ f(\mathbf{x}, \mathbf{y}) + \boldsymbol{\lambda}^\mathsf{T} g(\mathbf{x}, \mathbf{y}) \}$.

Although MP contains an infinite number of constraints, we can instead solve a Relaxed Master Problem (RMP):

$$\min_{\eta, \mathbf{y} \in \mathcal{Y}} \eta \qquad \text{s.\,t.} \ \ \eta \geq \inf_{\mathbf{x} \in \mathcal{X}} \left\{ f(\mathbf{x}, \mathbf{y}) + \boldsymbol{\lambda}^\mathsf{T} g(\mathbf{x}, \mathbf{y}) \right\}, \ \forall \boldsymbol{\lambda} \in \Lambda$$

where $\Lambda \subset \mathbb{R}^m$ consists of a finite set of dual variables. Let $(\eta^*, \mathbf{x}^*, \mathbf{y}^*)$ be an optimal solution to MP and let $(\hat{\eta}, \hat{\mathbf{y}})$ be an optimal solution to RMP. Then, $\hat{\eta} \leq \eta^*$ presents a lower bound to MP (and therefore problem (11)). Furthermore, $L(\hat{\mathbf{y}}) \geq f(\mathbf{x}^*, \mathbf{y}^*)$ gives an upper bound. Finally, let $\hat{\mathbf{x}}$ be the minimizer of $L(\hat{\mathbf{y}})$. Then $(\hat{\mathbf{x}}, \hat{\mathbf{y}})$ is a feasible solution achieving the above upper bound.

Carefully selecting $\Lambda$ will yield tighter relaxations to MP (Floudas & Pardalos, 2013). GBD proposes iteratively growing $\Lambda$ and solving increasingly larger RMPs until the upper and lower bounds converge. We initialize with a feasible $\hat{\mathbf{y}} \in \mathcal{Y}$ and $\Lambda = \emptyset$. Then in each iteration $t \in \{0, 1, 2, \dots\}$, we:

1. Given a solution $(\hat{\eta}^t, \hat{\mathbf{y}}^t)$, solve the Lagrangian $L(\hat{\mathbf{y}}^t)$ to obtain primal-dual solutions $(\hat{\mathbf{x}}^t, \hat{\boldsymbol{\lambda}}^t)$.

2. Update $\Lambda \leftarrow \Lambda \cup \{\hat{\boldsymbol{\lambda}}^t\}$ and solve RMP to obtain a solution $(\hat{\eta}^{t+1}, \hat{\mathbf{y}}^{t+1})$.

The algorithm terminates when $L(\hat{\mathbf{y}}^t) - \hat{\eta}^t$ is below a tolerance threshold $\varepsilon > 0$.

For many problems, the above algorithm will converge in a finite number of iterations. This is referred to as Property (P) in the literature (Geoffrion, 1972; Floudas & Pardalos, 2013).

**Lemma 2** (Geoffrion (1972)). *Consider the non-convex optimization problem (11). If $f(\mathbf{x}, \mathbf{y}) = f_1(\mathbf{x}) + f_2(\mathbf{y})$ and $g(\mathbf{x}, \mathbf{y}) = g_1(\mathbf{x}) + g_2(\mathbf{y})$ are separable, linear functions, then for any $\varepsilon > 0$, Generalized Benders Decomposition terminates in a finite number of iterations.*

Even so, the number of iterations may be exceedingly long. As a result, the modern literature in GBD explores new constraints and heuristics to quickly tighten the relaxation of RMP versus MP. Often, these techniques are customized to the structure of the specific optimization problem (11) being solved. In Section 3.2, we propose new custom constraints to improve the algorithm.

## D  DETAILS ON OUR GENERALIZED BENDERS DECOMPOSITION ALGORITHM FOR MINIMIZING WASSERSTEIN DISTANCE

In this section, we provide further details on the GBD algorithm used to solve problem (4). To complement the Enhanced Optimality Cuts in Section 3.2, we first discuss an additional set of pruning constraints to improve our algorithm. We then detail the full steps of the algorithm and provide Python-style pseudo-code.

### D.1  PRUNING CONSTRAINTS

We first define the pruning constraints used to improve our GBD algorithm. These constraints complement the Enhanced Optimality Cuts introduced in Section 3.2 as an additional set of constraints

---

**Algorithm 1** Generalized Benders Decomposition (GBD) for the Minimum Wasserstein Distance

---

1: **Input:** budget $B$; initial core set $\hat{\boldsymbol{\pi}}$; constraint sets $\Lambda = \emptyset$, $\mathcal{P}^+ = \{\hat{\boldsymbol{\pi}}\}$, $\mathcal{P}^- = \emptyset$; upper bound $\overline{B} = \infty$; lower bound $\underline{B} = -\infty$; hyperparameters $\beta^+$, $\beta^-$; tolerance $\varepsilon$; maximum time limit $T$
2: **repeat**
3:     Solve $W(\mathcal{C}(\hat{\boldsymbol{\pi}}), \mathcal{D})$: $\hat{\boldsymbol{\Gamma}} \leftarrow \arg\min$ Problem (1), $(\hat{\boldsymbol{\lambda}}, \hat{\boldsymbol{\mu}}) \leftarrow \arg\max$ Problem (2).
4:     $\Lambda \leftarrow \Lambda \cup \{(\hat{\boldsymbol{\pi}}, \hat{\boldsymbol{\Gamma}}, \hat{\boldsymbol{\lambda}}, \hat{\boldsymbol{\mu}})\}$           // Update GBD constraint set
5:     **if** $W(\mathcal{C}(\hat{\boldsymbol{\pi}}), \mathcal{D}) \leq \overline{B}$ **then**
6:         $\overline{B} \leftarrow \min(\overline{B}, W(\mathcal{C}(\hat{\boldsymbol{\pi}}), \mathcal{D}))$      // Update upper bound on optimal value
7:         $\mathcal{P}^+ \leftarrow \mathcal{P}^+ \cup \{\hat{\boldsymbol{\pi}}\}$         // Update Near Neighbourhood set
8:     **else**
9:         $\mathcal{P}^- \leftarrow \mathcal{P}^- \cup \{\hat{\boldsymbol{\pi}}\}$         // Update Prune Neighbourhood set
10:     **end if**
11:     Solve W-RMP($\Lambda$):

$$(\hat{\eta}, \hat{\boldsymbol{\pi}}) \leftarrow \arg\min_{\eta, \boldsymbol{\pi} \in \mathcal{P}} \quad \eta$$

$$\text{s.t.} \quad \eta \geq \frac{1}{B} \hat{\boldsymbol{\lambda}}^{\mathsf{T}} \left( \boldsymbol{\pi} - \frac{1}{B} \underbrace{\hat{\boldsymbol{\Gamma}}^{\mathsf{T}} \mathbf{1}}_{=\hat{\boldsymbol{\pi}}} \right) + W(\mathcal{C}(\hat{\boldsymbol{\pi}}), \mathcal{D}), \quad \forall (\hat{\boldsymbol{\pi}}, \hat{\boldsymbol{\Gamma}}, \hat{\boldsymbol{\lambda}}, \hat{\boldsymbol{\mu}}) \in \Lambda$$

        Lower Bound, Eq. (6)

        Triangle Ineq., Eq. (7),             $\forall (\hat{\boldsymbol{\pi}}, \hat{\boldsymbol{\Gamma}}, \hat{\boldsymbol{\lambda}}, \hat{\boldsymbol{\mu}}) \in \Lambda$

        Dual Ineq., Eq. (8),                $\forall (\hat{\boldsymbol{\pi}}, \hat{\boldsymbol{\Gamma}}, \hat{\boldsymbol{\lambda}}, \hat{\boldsymbol{\mu}}) \in \Lambda$

        $\boldsymbol{\pi}^{\mathsf{T}} \hat{\boldsymbol{\pi}} \geq \beta^+ B,$                 $\forall \hat{\boldsymbol{\pi}} \in \mathcal{P}^+$

        $\boldsymbol{\pi}^{\mathsf{T}} \hat{\boldsymbol{\pi}} \leq \beta^- B,$                 $\forall \hat{\boldsymbol{\pi}} \in \mathcal{P}^-$

12:     $\underline{B} \leftarrow \max(\underline{B}, \hat{\eta})$         // Update lower bound on optimal value
13: **until** $T$ hours or $\overline{B} - \underline{B} < \varepsilon$         // Exit at timeout or convergence

---

used to accelerate the algorithm. However, these pruning constraints do not preserve the convergence criteria of Corollary 1, since there is no guarantee that the optimal solution will satisfy them. Instead, these constraints are heuristics that our numerical results empirically show to improve the active learning strategy.

Let $\mathcal{P}^+$ and $\mathcal{P}^-$ denote two sets of good and bad candidate core sets $\hat{\boldsymbol{\pi}}$, respectively, and let $\beta^+, \beta^- \in (0, 1)$ denote two hyperparameters. In each iteration, we obtain a candidate $\hat{\boldsymbol{\pi}}$ and a corresponding upper bound $W(\mathcal{C}(\hat{\boldsymbol{\pi}}), \mathcal{D})$ to the W-MP. If this upper bound is a new incumbent (i.e., is lower than the previous upper bounds) then we update $\mathcal{P}^+ \cup \{\hat{\boldsymbol{\pi}}\}$ and limit the search to within a Hamming ball of size $\beta^+ B$ around $\hat{\boldsymbol{\pi}}$. Otherwise, we update $\mathcal{P}^- \cup \{\hat{\boldsymbol{\pi}}\}$ and remove a Hamming ball of size $\beta^- B$ around $\hat{\boldsymbol{\pi}}$ from the search space. These two pruning techniques can be described by the following linear constraints: $\forall \hat{\boldsymbol{\pi}} \in \mathcal{P}^+, \ \boldsymbol{\pi}^{\mathsf{T}} \hat{\boldsymbol{\pi}} \geq \beta^+ B$ and $\forall \hat{\boldsymbol{\pi}} \in \mathcal{P}^-, \ \boldsymbol{\pi}^{\mathsf{T}} \hat{\boldsymbol{\pi}} \leq \beta^- B$.

### D.2 ALGORITHM STEPS AND PSEUDO-CODE

Algorithm 1 details the mathematical steps of the GBD algorithm and Figure 5 provides a Python-style pseudocode snippet. We formulate our algorithm in a loop by iteratively solving our optimization problem and adding new constraints. The main dependency of this optimization algorithm lies in an off-the-shelf Wasserstein distance computation library (Flamary et al., 2021) and a MILP solver (Forrest & Lougee-Heimer, 2005). Note that rather than running GBD for a fixed number of iterations, Algorithm 1 performs iterations until a maximum time limit $T$ or until the upper $\overline{B}$ and lower bounds $\underline{B}$ converge under a tolerance $\varepsilon$. Furthermore in the implementation of the algorithm, we may also select hyperparameters for the MILP solver being used to solve each instance of W-RMP($\Lambda$). For our experiments, we only tune the time limit and the minimum optimality gap. That is, we set the solver for W-RMP($\Lambda$) to timeout in 180 seconds or until the solution it finds is within 10% of optimality.

```python
# Input
features = ...  # Array of feature vectors
N, B = ...  # Total number of points and budget
prev_selected = ... # Array of indices selected in earlier rounds

# Calculate distance matrix
arcs = pairwise_distances(features)

# Define W-RMP MIP model and variables
prob = mip.Model()
pis = [prob.add_var(var_type=mip.BINARY) for i in range(N)]
eta = prob.add_var()

# Set objective and constraints
prob.objective = mip.minimize(eta)
prob += mip.xsum(pis) == B

# If we labeled a point in an earlier round, then it should be 1
for i in prev_selected:
    prob += pis[i] == 1

# GBD loop
lb, ub, tol, max_time = -inf, inf, 1e-3, 10800
while lb < ub - tol and current_time() < max_time:
    # Run MIP solver to populate pis, eta
    prob.optimize()

    # Compute Wasserstein distance and dual variables with POT
    pis_x = [p.x for p in pis]
    w_hat, info = ot.emd2(pis_x / N, ones(N) / N, arcs, log=True)
    lambda_hat, mu_hat = info['u'], info['v']

    # Update bounds
    lb, ub = max(lb, eta.x), min(ub, w_hat)

    # Add Benders cut
    prob += eta >= w_hat + mip.xsum(
        lambda_hat[i] * (pis[i] - pis.x[i]) / B for i in range(N))

    # Triangle inequality cut
    min_Di = ...  # Compute coefficients according to Proposition 1
    prob += eta + w_hat >= mip.xsum(
        min_Di[i] * pis[i] / B for i in range(N))

    # Dual inequality cut
    max_Di = ...  # Compute coefficients according to Proposition 1
    prob += eta >= w_hat + mip.xsum(
        (lambda_hat[i] - max_Di[i]) * pis[i] / B - mu_hat[i] / N
        for i in range(N))

    # Pruning constraints
    if ub == w_hat:
        prob += B * beta_plus <= mip.xsum(
            pis[i] * pis.x[i] for i in range(N))
    else:
        prob += B * beta_minus >= mip.xsum(
            pis[i] * pis.x[i] for i in range(N))

# Output
return where(pis.x == 1)  # All selected indices so far
```

Figure 5: Pseudocode of the GBD algorithm used to solve our optimization problem. The above code uses the Python-MIP (`mip`) and Python Optimal Transport (`ot`) libraries.)

### D.3 REDUCING THE COMPUTATIONAL COMPLEXITY

The original problem (4) is a MILP with $N^2 + N$ variables and $2N + 1$ constraints. This problem is too large to solve on standard computers when $N \geq 50,000$, which is typical for deep learning data sets. Instead in Algorithm 1, we repeatedly solve two smaller sub-problems: (i) W-RMP($\Lambda$) to obtain selections $\hat{\pi}$ and (ii) the Wasserstein distance $W(\mathcal{C}(\hat{\pi}), \mathcal{D})$ to obtain dual variables $\hat{\lambda}$.

W-RMP($\Lambda$) is a MILP with $N + 1$ variables and $|\Lambda| + 1$ constraints. Incorporating the Enhanced Optimality Cuts and the Pruning Constraints leads to a MILP with $N + 1$ variables and $5|\Lambda| + 1$ constraints. From our experiments, $|\Lambda|$ is typically on the order of 100, meaning that this MILP contains several orders of magnitude fewer variables and constraints than problem (4). We can find an optimal or nearly optimal solution to this MILP in less than 180 seconds for most settings.

Before computing the discrete Wasserstein distance $W(\mathcal{C}(\hat{\pi}), \mathcal{D})$, we first need the distance matrix of features $\mathbf{D_x}$. For Euclidean distance, computing this has complexity $O(dN^2)$ where $d$ is the feature dimension. Note that this is a one-time cost and that once we construct the distance matrix, we can store it in memory for the remainder of the algorithm.

To compute $W(\mathcal{C}(\hat{\pi}), \mathcal{D})$ in each iteration, we use an off-the-shelf solver, POT (Flamary et al., 2021). POT applies the Network Simplex Algorithm (Tarjan, 1997), which is an algorithm for solving network flow problems over graphs. Because $W(\mathcal{C}(\hat{\pi}), \mathcal{D})$ is a minimum flow over a graph with $2N$ nodes and $N^2$ edges, our complexity is $O(N^3 (\log N)^2)$.

When $B$ is much smaller than $N$, we can often simply approximate $\hat{\lambda}$. We first remove nodes corresponding to $\hat{\pi}_i = 0$ to rewrite $W(\mathcal{C}(\hat{\pi}), \mathcal{D})$ as an equivalent network flow with $N + B$ nodes and $NB$ edges, resulting in complexity $O(N^2 (\log N)^2)$. However, from (2), this problem returns dual variables corresponding only to the $B$ points for which $\hat{\pi}_i = 1$. We can find the remaining dual variables via a c-transform (Peyré et al., 2019), which requires solving a minimization problem over an $N$-length array for each remaining variable. We refer the reader to Peyré et al. (2019) for details. This approach does not guarantee an optimal $\hat{\lambda}$, but we find that it works well in practice.

Rather than exact algorithms, we could also approximate $W(\mathcal{C}(\hat{\pi}), \mathcal{D})$ with Sinkhorn distances (Cuturi, 2013; Peyré et al., 2019). These can typically be computed faster than exact approaches and can also be scaled to large $N$ with further approximations (Altschuler et al., 2019). Moreover, Sinkhorn distances can be slotted directly into the vanilla GBD algorithm by replacing the dual regularizer $\hat{\lambda}$ with the gradient of the Sinkhorn distance in the Benders constraints (Cuturi & Doucet, 2014; Luise et al., 2018). However, they may not necessarily satisfy the EOC in Proposition 1 and thereby prevent further accelerations.

## E COMPLETE NUMERICAL RESULTS

This section expands the numerical results summarized in the main paper. Below, we summarize the main points:

- Section E.1 details the hyperparameters that were used in our GBD algorithm and optimization solver. We also describe model choices and training settings used in our image classification and domain adaptation experiments.

- Section E.2 describes our experiments on optimizing problem (4). We first summarize the related scenario reduction literature and highlight the $k$-Medoids Cluster Center algorithm baseline. We then expand the numerical results from the main paper to show that for generating high-quality solutions to problem (4), our approach typically outperforms $k$-medoids and sometimes by up to $2\times$. Furthermore, these improvements remain even when we limit our algorithm to comparable runtimes with $k$-medoids.

- Section E.3 provides the full table of values for active learning experiments on the three image classification data sets. In addition to the results in the main paper, this section shows that our approach is consistently effective even up to budgets of $B = 180$.

- Section E.4 provides the full table of values for active learning with domain adaptation experiments. We evaluate all six source-target combinations on the Office-31 data set to show our approach either outperforms or competitive with the best baseline. To further

compare against $k$-centers, we perform statistical $t$-tests in a head-to-head setting. Our approach statistically outperforms $k$-centers for more budgets and data sets.

- Section E.5 details our ablation on classical active learning without self-supervised features. We first describe the experiment setup and then expand on the results in the main paper, showing that our algorithm is effective even without self-supervised features and in higher budgets.

- Section E.6 provides the full table of selection runtimes. Most baselines require at most a few minutes to select points. We further discuss the downstream effect of our algorithm having longer runtimes in practice.

- Section E.7 highlights two additional results not discussed in the main paper. We first ablate the value of customizing the base metric in the Wasserstein distance by showing that Cosine distance yields significant improvements over Euclidean distance on the CIFAR-10 data set. We then provide a qualitative analysis of our selections by plotting $t$-SNE visualizations of the feature space as well as samples of images selected. In general, our visualizations show that we can better cover the feature space than baselines.

## E.1 METHODS

We first detail hyperparameters for the optimization algorithm, methods for the image classification experiments, and methods for the domain adaptation experiments. All experiments are performed on computers with a single NVIDIA V100 GPU card and 50 GB CPU memory. Our optimization problem is solved on CPU.

**Algorithm Settings.** When formulating the Wasserstein distance problem, we use Cosine distance (i.e., $l_2$ normalize our features) as the base metric for the image classification experiments. Cosine distance is a better choice than Euclidean distance here because the features are obtained with SimCLR, which involves maximizing a Cosine distance between different images; that is, the optimization and feature generation use the same base metric. We include ablation tests in Appendix E.7 comparing Cosine versus Euclidean distance on CIFAR-10. We test both Cosine and Euclidean distance for the domain adaptation experiments. These experiments use features obtained via $f$-DAL, which is not necessarily tied to Cosine distance.

We implement the optimization problem using the COIN-OR/CBC Solver (Forrest & Lougee-Heimer, 2005). Since it is crucial to run as many iterations as possible to obtain better solutions, we set the solver optimality gap tolerance to 10% and a time limit of 180 seconds to solve W-RMP($\Lambda$) in each iteration. We also warm-start our algorithm with the solution to the $k$-centers core set, i.e., $\hat{\pi}_0 = \hat{\pi}_{k\text{-centers}}$, for image classification and $k$-centers or $k$-medoids for domain adaptation. We run Algorithm 1 with $\varepsilon = 10^{-3}$ and $T = 3$ hours for STL-10, CIFAR-10, and Office-31, and $T = 6$ hours for SVHN, which contains a larger data set. Finally for Wass. + EOC + P, we set $(\beta^+, \beta^-) = (0.6, 0.99)$. These are only optimization parameters and can be tuned by evaluating the objective function value of the optimization problem.

**Image Classification.** Table 5 details the differences in setup for each of the three data sets used in the image classification experiments. For each experiment, we use a ResNet feature encoder with a two-layer MLP (Linear-BatchNorm-ReLU-Linear) head. The first linear layer has a 64-dimension output in order to reduce the dimensions of the feature space. We use a 64-dimension second layer in the SimCLR pre-training step and a 10-dimension second layer during supervised learning. The choice of specific ResNet encoder was made based on the computational requirement of training on a single GPU under the required batch sizes.

For each experiment, we use the default test sets to evaluate our strategies. In the SimCLR representation learning step, we train for 800 epochs with an initial learning rate of 0.1 decaying to 0.001. We implement SimCLR following the steps of Chen et al. (2020c) with the following random data augmentations in their default PyTorch settings: resized crops, horizontal flips, color jitter, and grayscale. For SVHN, we omit the horizontal flips and increase the scale of the crops to $(0.8, 1)$, because this yielded better representations in our experiments. In the supervised learning step, we train with cross entropy loss, random crops and horizontal flips, and a batch size of 128 with a constant learning rate of 0.002 for 1600 epochs.

Table 5: Summary of our setup in the image classification experiments. *In each of the first two iterations of each data set, we use only half of the labeling budget.

| Dataset | Unlabled pool | Budget per iteration* | Model | SimCLR batch size | Algorithm runtime |
|---|---|---|---|---|---|
| STL10 | $5,000$ | 20 | ResNet-18 | 128 | 3 hrs. |
| CIFAR10 | $50,000$ | 20 | ResNet-34 | 512 | 3 hrs. |
| SVHN | $73,257$ | 20 | ResNet-34 | 512 | 6 hrs. |

Table 6: Summary of our setup in the domain adaptation experiments. *For each experiment, we randomly split the target set to $70\%$ training and $30\%$ testing sets.

| Task | Unlabled pool* | Budget per iteration | Initial solution | Base metric |
|---|---|---|---|---|
| DSLR to Amazon | $1,972$ | 50 | $k$-centers | Euclidean |
| Web to Amazon | $1,972$ | 50 | $k$-medoids | Euclidean |
| Web to DSLR | 349 | 10 | $k$-centers | Cosine |
| Amazon to DSLR | 349 | 10 | $k$-medoids | Euclidean |
| DSLR to Web | 557 | 10 | $k$-medoids | Cosine |
| Amazon to Web | 557 | 10 | $k$-centers | Cosine |

**Domain Adaptation.** Table 6 details the differences in setup for each of the tasks in the domain adaptation experiment. We use a ResNet-50 + one-layer bottleneck (Linear-BatchNorm-ReLU-Dropout) encoder and a two-layer classification head (Linear-ReLU-Dropout-Linear). The ResNet-50 is initialized with Imagenet-pretrained weights. The bottleneck sets the feature space dimension to $1,000$. The second layer of the classification head reduces the dimension to 31 for the number of classes.

For each experiment, we randomly partition the target data set into $70\%$ for training and $30\%$ for testing. We perform the representation learning step using the full labeled source data set and the unlabeled target training data set. We then perform the supervised learning step using only labeled target training data. The representation learning is performed via the unsupervised domain adaptation algorithm, $f$-DAL (Acuna et al., 2021). We train for 20 epochs with $1,000$ iterations per epoch. We use a learning rate of $0.004$ decaying stepwise and a batch size of 32, which are the default parameters of Acuna et al. (2021). In the supervised learning step, we train with cross entropy loss with the same learning rate and batch size.

### E.2 OBJECTIVE FUNCTION VALUE

We evaluate GBD on minimizing the Wasserstein distance by comparing it to the $k$-Medoids Cluster Center algorithm from scenario reduction for stochastic optimization. We first briefly summarize the scenario reduction problem. We then expand the numerical results in the main paper, showing that for large problem sizes, our algorithm yields better minimizers of the Wasserstein distance.

**Background.** Consider a general stochastic optimization problem

$$\min_{\mathbf{w} \in \mathcal{W}} \ \mathbb{E}_{\mathbf{x} \sim \mathbb{P}}[f(\mathbf{x}, \mathbf{w})]$$

where $f(\mathbf{x}, \mathbf{w})$ is an objective function, $\mathbf{w} \in \mathcal{W}$ is the variable to optimize and $\mathbf{x} \sim \mathbb{P}$ is a random variable, here referred to as a scenario. Given a data set $\mathcal{D} := \{\hat{\mathbf{x}}_i\}_{i=1}^{N}$ of $N$ scenarios, we can solve the Sample Average Approximation (SAA), which corresponds to minimizing the empirical expected value. Machine learning is a special case of the above problem where $f(\mathbf{x}, \mathbf{w})$ corresponds to the loss function, $\mathbf{w}$ to neural network weights, and $\mathbf{x}$ to training data.

SAA may grow difficult to solve with large data sets, especially if $\mathcal{W}$ is non-convex or contains integral constraints. Here, scenario reduction is the problem of formulating an approximation of the SAA by using only a subset $\mathcal{C} := \{\hat{\mathbf{x}}_j\}_{j=1}^{B} \subset \mathcal{D}$ of $B$ scenarios. Heitsch & Römisch (2003) consider constructing this subset by minimizing the Wasserstein distance between $\mathcal{C}$ and $\mathcal{D}$, i.e., by solving problem (4). When $N$ is large, Heitsch & Römisch (2003) propose the $k$-Medoids Cluster Center algorithm as a heuristic. Although the algorithm is computationally easy to implement, making it

Table 7: Head-to-head comparison of our solver versus $k$-medoids on objective function value of problem (4) at different budgets. For each data set, the best solution for each budget is bolded and underlined. All entries show mean $\pm$ standard deviation over five runs.

| $B$ | STL-10 | | CIFAR-10 | | SVHN | |
|---|---|---|---|---|---|---|
| | $k$-medoids | Wass. + EOC | $k$-medoids | Wass. + EOC | $k$-medoids | Wass. + EOC |
| 10 | $0.290 \pm 0.028$ | $\mathbf{0.281 \pm 0.005}$ | $\mathbf{0.258 \pm 0.013}$ | $0.322 \pm 0.012$ | $\mathbf{0.152 \pm 0.009}$ | $0.153 \pm 0.007$ |
| 20 | $0.132 \pm 0.015$ | $\mathbf{0.134 \pm 0.007}$ | $\mathbf{0.142 \pm 0.013}$ | $0.176 \pm 0.010$ | $0.172 \pm 0.010$ | $\mathbf{0.093 \pm 0.010}$ |
| 40 | $\mathbf{0.109 \pm 0.005}$ | $0.122 \pm 0.007$ | $\mathbf{0.118 \pm 0.005}$ | $0.125 \pm 0.006$ | $0.164 \pm 0.005$ | $\mathbf{0.093 \pm 0.008}$ |
| 60 | $\mathbf{0.110 \pm 0.006}$ | $0.116 \pm 0.003$ | $0.120 \pm 0.005$ | $\mathbf{0.092 \pm 0.006}$ | $0.175 \pm 0.005$ | $\mathbf{0.100 \pm 0.006}$ |
| 80 | $\mathbf{0.100 \pm 0.007}$ | $0.101 \pm 0.002$ | $0.109 \pm 0.003$ | $\mathbf{0.077 \pm 0.004}$ | $0.178 \pm 0.003$ | $\mathbf{0.098 \pm 0.007}$ |
| 100 | $0.097 \pm 0.003$ | $\mathbf{0.096 \pm 0.002}$ | $0.104 \pm 0.006$ | $\mathbf{0.097 \pm 0.003}$ | $0.179 \pm 0.006$ | $\mathbf{0.091 \pm 0.006}$ |
| 120 | $\mathbf{0.092 \pm 0.004}$ | $\mathbf{0.092 \pm 0.004}$ | $0.095 \pm 0.007$ | $\mathbf{0.064 \pm 0.006}$ | $0.178 \pm 0.004$ | $\mathbf{0.085 \pm 0.004}$ |
| 140 | $0.089 \pm 0.004$ | $\mathbf{0.087 \pm 0.004}$ | $0.089 \pm 0.007$ | $\mathbf{0.059 \pm 0.002}$ | $0.173 \pm 0.004$ | $\mathbf{0.080 \pm 0.004}$ |
| 160 | $0.093 \pm 0.005$ | $\mathbf{0.090 \pm 0.004}$ | $0.099 \pm 0.005$ | $\mathbf{0.072 \pm 0.005}$ | $0.173 \pm 0.005$ | $\mathbf{0.087 \pm 0.004}$ |
| 180 | $0.087 \pm 0.005$ | $\mathbf{0.085 \pm 0.002}$ | $0.096 \pm 0.004$ | $\mathbf{0.072 \pm 0.003}$ | $0.170 \pm 0.005$ | $\mathbf{0.089 \pm 0.004}$ |

Table 8: Head-to-head comparison of Wass. + EOC with different runtimes versus $k$-medoids on objective function value of problem (4) for the CIFAR-10 data set at different budgets. The best solution for each budget is bolded and underlined and the second best solution is underlined.

| $B$ | $k$-medoids | Wass. + EOC | | | | |
|---|---|---|---|---|---|---|
| | | 3 min. | 6 min. | 20 min. | 1 hr. | 3 hr. |
| 10 | $\mathbf{0.258 \pm 0.013}$ | $0.352 \pm 0.010$ | $0.328 \pm 0.010$ | $0.337 \pm 0.008$ | $0.326 \pm 0.007$ | $0.322 \pm 0.012$ |
| 20 | $\mathbf{0.142 \pm 0.013}$ | $0.198 \pm 0.012$ | $0.184 \pm 0.003$ | $0.178 \pm 0.008$ | $0.178 \pm 0.008$ | $0.176 \pm 0.010$ |
| 40 | $\mathbf{0.118 \pm 0.005}$ | $0.159 \pm 0.010$ | $0.131 \pm 0.008$ | $0.126 \pm 0.003$ | $0.122 \pm 0.007$ | $0.125 \pm 0.006$ |
| 60 | $0.120 \pm 0.005$ | $0.118 \pm 0.016$ | $0.103 \pm 0.006$ | $0.102 \pm 0.010$ | $0.089 \pm 0.007$ | $\mathbf{0.092 \pm 0.006}$ |
| 80 | $0.109 \pm 0.003$ | $0.102 \pm 0.020$ | $0.091 \pm 0.008$ | $0.084 \pm 0.008$ | $0.078 \pm 0.002$ | $\mathbf{0.077 \pm 0.004}$ |
| 100 | $0.104 \pm 0.006$ | $0.088 \pm 0.009$ | $0.077 \pm 0.003$ | $0.076 \pm 0.007$ | $\mathbf{0.071 \pm 0.002}$ | $0.097 \pm 0.003$ |
| 120 | $0.095 \pm 0.007$ | $0.082 \pm 0.006$ | $0.072 \pm 0.006$ | $0.068 \pm 0.004$ | $0.065 \pm 0.003$ | $\mathbf{0.064 \pm 0.006}$ |
| 140 | $0.089 \pm 0.007$ | $0.073 \pm 0.004$ | $0.068 \pm 0.001$ | $0.064 \pm 0.004$ | $0.061 \pm 0.004$ | $\mathbf{0.059 \pm 0.002}$ |
| 160 | $0.099 \pm 0.005$ | $0.084 \pm 0.005$ | $0.079 \pm 0.002$ | $0.076 \pm 0.003$ | $0.075 \pm 0.004$ | $\mathbf{0.072 \pm 0.005}$ |
| 180 | $0.096 \pm 0.004$ | $0.080 \pm 0.003$ | $0.075 \pm 0.005$ | $0.072 \pm 0.002$ | $0.072 \pm 0.002$ | $\mathbf{0.072 \pm 0.003}$ |

desirable in practice, recently Rujeerapaiboon et al. (2018) show that $k$-medoids can potentially be a poor approximation in the worst case, and also introduce a new approximation algorithm.

Finally, we note that the scenario reduction literature rarely considers problem sizes of the scale in our active learning tasks, with numerical experiments in most cases ranging up to $10^3$. In contrast, the SVHN data set requires $N \leq 75,000$.

**Comparing the Minimum Wasserstein Distance.** Our GBD algorithm directly solves the integer optimization problem (4) rather than a heuristic or approximation algorithm. We compare against $k$-medoids on minimizing the Wasserstein distance for the three image classification data sets.

Table 7 presents the complete results of our method versus $k$-medoids for each data set. We run the solver for three hours for STL-10 and CIFAR-10 and for six hours for SVHN. Although the algorithm has not yet converged, for each data set, our approach is competitive for all settings (except for $B \leq 20$ at CIFAR-10), and especially outperforms $k$-medoids for $B \geq 120$. Finally, note that for SVHN, we reduce the objective function value by over half compared to $k$-medoids.

To further ablate our method, we also compare different runtimes of CIFAR-10 with $k$-medoids in Table 8. Even down to three minutes, we often get a better minimizer than $k$-medoids. Note from Table 12 that $k$-medoids requires on average several minutes to compute a selection. This suggests that our algorithm may have more general use cases outside of active learning, such as an alternative to $k$-medoids in scenario reduction.

### E.3 IMAGE CLASSIFICATION

Table 9 lists the means $\pm$ standard deviations of accuracies shown in Figure 2. Our models consistently outperform the baselines for both STL-10 and SVHN, with the exception of the first round in STL-10. On CIFAR-10, our models outperform in the first two rounds, but afterwards remain competitive with $k$-centers (i.e., less than 2% difference in accuracy). On SVHN, our models sim-

Table 9: Numerical values of main results in Figure 2. All entries show mean $\pm$ standard deviation over five runs. The best model for each budget range is bolded and underlined and the second best model is underlined.

| | | | | STL-10 | | | | |
|---|---|---|---|---|---|---|---|---|
| $B$ | Random | Confidence | Entropy | $k$-medoids | $k$-centers | WAAL | Wass. + EOC | Wass. + EOC + P |
| 10 | $30.2 \pm 6.1$ | $30.2 \pm 6.1$ | $30.2 \pm 6.1$ | $\mathbf{45.7 \pm 3.5}$ | $37.4 \pm 4.4$ | $30.2 \pm 6.1$ | $43.1 \pm 0.7$ | $41.4 \pm 3.2$ |
| 20 | $42.8 \pm 3.6$ | $37.8 \pm 4.2$ | $33.6 \pm 4.1$ | $50.2 \pm 2.4$ | $45.3 \pm 4.6$ | $45.8 \pm 4.0$ | $\mathbf{51.5 \pm 2.1}$ | $49.5 \pm 3.6$ |
| 40 | $51.0 \pm 3.4$ | $46.6 \pm 3.1$ | $48.5 \pm 3.4$ | $56.5 \pm 1.2$ | $53.5 \pm 2.9$ | $53.6 \pm 4.2$ | $\mathbf{60.4 \pm 1.5}$ | $58.7 \pm 1.8$ |
| 60 | $59.1 \pm 1.6$ | $55.0 \pm 4.1$ | $57.2 \pm 1.6$ | $59.5 \pm 1.8$ | $60.0 \pm 1.6$ | $59.7 \pm 3.0$ | $62.9 \pm 4.3$ | $\mathbf{63.8 \pm 1.6}$ |
| 80 | $62.7 \pm 1.3$ | $59.9 \pm 3.1$ | $60.0 \pm 2.2$ | $62.2 \pm 2.3$ | $64.1 \pm 1.5$ | $60.8 \pm 1.7$ | $\mathbf{67.9 \pm 1.1}$ | $66.5 \pm 1.2$ |
| 100 | $65.4 \pm 1.1$ | $63.4 \pm 3.0$ | $63.1 \pm 2.2$ | $64.3 \pm 1.9$ | $66.2 \pm 1.7$ | $62.4 \pm 1.8$ | $69.3 \pm 0.4$ | $\mathbf{70.8 \pm 1.4}$ |
| 120 | $67.9 \pm 1.1$ | $66.8 \pm 1.8$ | $66.6 \pm 1.3$ | $66.3 \pm 1.4$ | $68.7 \pm 2.0$ | $64.3 \pm 1.5$ | $71.0 \pm 1.4$ | $\mathbf{73.3 \pm 0.6}$ |
| 140 | $70.8 \pm 0.6$ | $69.1 \pm 2.2$ | $70.0 \pm 0.6$ | $69.2 \pm 1.0$ | $71.9 \pm 1.4$ | $66.7 \pm 1.8$ | $73.0 \pm 1.3$ | $\mathbf{73.4 \pm 0.5}$ |
| 160 | $71.1 \pm 0.5$ | $69.4 \pm 2.5$ | $70.4 \pm 0.8$ | $68.5 \pm 1.7$ | $72.0 \pm 1.5$ | $67.7 \pm 2.0$ | $73.4 \pm 1.5$ | $\mathbf{73.6 \pm 0.5}$ |
| 180 | $71.3 \pm 1.1$ | $71.0 \pm 1.3$ | $71.2 \pm 0.6$ | $69.6 \pm 1.0$ | $71.1 \pm 0$ | $68.6 \pm 1.7$ | $\mathbf{73.7 \pm 1.1}$ | $\mathbf{73.7 \pm 0.9}$ |
| | | | | CIFAR-10 | | | | |
| $B$ | Random | Confidence | Entropy | $k$-medoids | $k$-centers | WAAL | Wass. + EOC | Wass. + EOC + P |
| 10 | $38.0 \pm 9.7$ | $38.0 \pm 9.7$ | $38.0 \pm 9.7$ | $45.6 \pm 5.9$ | $37.0 \pm 3.2$ | $38.0 \pm 9.7$ | $\mathbf{50.7 \pm 3.0}$ | $46.8 \pm 1.6$ |
| 20 | $51.8 \pm 8.5$ | $37.7 \pm 4.5$ | $34.4 \pm 3.1$ | $53.7 \pm 7.0$ | $56.2 \pm 5.9$ | $40.9 \pm 5.5$ | $63.7 \pm 4.8$ | $\mathbf{65.3 \pm 2.9}$ |
| 40 | $65.1 \pm 3.1$ | $47.8 \pm 3.1$ | $40.5 \pm 6.3$ | $65.3 \pm 3.0$ | $\mathbf{74.7 \pm 4.3}$ | $50.1 \pm 3.3$ | $73.5 \pm 1.6$ | $72.7 \pm 1.8$ |
| 60 | $70.1 \pm 3.6$ | $55.3 \pm 5.8$ | $50.0 \pm 7.9$ | $69.4 \pm 3.2$ | $\mathbf{78.9 \pm 0.9}$ | $59.3 \pm 5.5$ | $77.6 \pm 1.3$ | $78.4 \pm 1.0$ |
| 80 | $74.5 \pm 3.9$ | $60.9 \pm 5.6$ | $58.6 \pm 7.2$ | $71.0 \pm 1.5$ | $\mathbf{81.0 \pm 1.2}$ | $62.7 \pm 5.8$ | $80.3 \pm 0.9$ | $79.5 \pm 1.2$ |
| 100 | $74.6 \pm 3.6$ | $67.8 \pm 2.8$ | $62.8 \pm 6.6$ | $72.6 \pm 0.9$ | $81.8 \pm 0.6$ | $64.4 \pm 5.8$ | $81.2 \pm 1.4$ | $81.5 \pm 0.7$ |
| 120 | $78.2 \pm 2.4$ | $71.0 \pm 2.2$ | $66.4 \pm 4.4$ | $74.0 \pm 0.8$ | $82.3 \pm 0.7$ | $65.7 \pm 3.8$ | $\mathbf{82.4 \pm 0.6}$ | $82.1 \pm 0.5$ |
| 140 | $80.1 \pm 1.9$ | $74.2 \pm 3.1$ | $71.4 \pm 4.6$ | $74.9 \pm 0.4$ | $83.0 \pm 0.6$ | $69.0 \pm 1.9$ | $82.9 \pm 1.0$ | $\mathbf{83.2 \pm 0.6}$ |
| 160 | $81.0 \pm 1.9$ | $75.9 \pm 3.6$ | $74.0 \pm 3.7$ | $75.9 \pm 0.8$ | $83.0 \pm 0.7$ | $69.0 \pm 1.4$ | $\mathbf{83.3 \pm 0.7}$ | $83.2 \pm 0.9$ |
| 180 | $82.0 \pm 1.3$ | $74.9 \pm 2.8$ | $74.9 \pm 3.5$ | $76.8 \pm 1.5$ | $\mathbf{83.7 \pm 0.6}$ | $68.9 \pm 1.5$ | $83.4 \pm 0.6$ | $83.4 \pm 1.1$ |
| | | | | SVHN | | | | |
| $B$ | Random | Confidence | Entropy | $k$-medoids | $k$-centers | WAAL | Wass. + EOC | Wass. + EOC + P |
| 10 | $37.1 \pm 4.9$ | $37.1 \pm 4.9$ | $37.1 \pm 4.9$ | $35.7 \pm 6.7$ | $28.6 \pm 4.6$ | $37.1 \pm 4.9$ | $37.0 \pm 2.8$ | $\mathbf{38.0 \pm 2.9}$ |
| 20 | $47.2 \pm 4.2$ | $36.9 \pm 4.1$ | $41.4 \pm 4.1$ | $46.1 \pm 4.6$ | $32.6 \pm 6.4$ | $41.5 \pm 2.3$ | $48.3 \pm 3.6$ | $\mathbf{49.4 \pm 4.0}$ |
| 40 | $57.5 \pm 6.3$ | $48.3 \pm 4.3$ | $51.0 \pm 2.8$ | $58.6 \pm 6.3$ | $54.2 \pm 4.4$ | $39.8 \pm 3.6$ | $63.2 \pm 6.5$ | $\mathbf{63.8 \pm 6.1}$ |
| 60 | $63.8 \pm 5.3$ | $50.7 \pm 5.9$ | $56.1 \pm 2.8$ | $65.4 \pm 6.0$ | $68.9 \pm 4.9$ | $47.8 \pm 4.1$ | $69.4 \pm 4.6$ | $\mathbf{72.1 \pm 6.6}$ |
| 80 | $64.7 \pm 3.7$ | $55.6 \pm 2.9$ | $59.7 \pm 3.3$ | $70.7 \pm 3.0$ | $\mathbf{78.5 \pm 2.8}$ | $51.4 \pm 4.1$ | $73.1 \pm 2.5$ | $77.4 \pm 3.9$ |
| 100 | $68.1 \pm 2.8$ | $58.5 \pm 3.3$ | $64.3 \pm 2.0$ | $73.6 \pm 2.6$ | $\mathbf{82.8 \pm 2.2}$ | $52.3 \pm 2.9$ | $77.6 \pm 3.1$ | $80.5 \pm 3.1$ |
| 120 | $72.1 \pm 1.8$ | $59.9 \pm 3.0$ | $67.2 \pm 1.4$ | $77.5 \pm 2.5$ | $\mathbf{84.3 \pm 1.7}$ | $54.9 \pm 2.8$ | $81.6 \pm 3.5$ | $82.9 \pm 3.3$ |
| 140 | $78.8 \pm 2.1$ | $65.9 \pm 4.4$ | $73.4 \pm 1.1$ | $82.0 \pm 3.0$ | $87.3 \pm 1.6$ | $59.5 \pm 3.1$ | $85.9 \pm 3.2$ | $\mathbf{87.6 \pm 2.6}$ |
| 160 | $80.4 \pm 1.6$ | $68.8 \pm 4.0$ | $73.6 \pm 1.5$ | $82.2 \pm 1.9$ | $\mathbf{87.6 \pm 1.9}$ | $58.2 \pm 1.9$ | $86.9 \pm 2.6$ | $\mathbf{87.6 \pm 1.8}$ |
| 180 | $81.2 \pm 1.1$ | $69.9 \pm 2.5$ | $74.2 \pm 1.7$ | $83.3 \pm 2.2$ | $87.3 \pm 1.4$ | $62.1 \pm 4.0$ | $86.7 \pm 2.6$ | $\mathbf{88.6 \pm 1.2}$ |

ilarly outperform in the first four rounds at which point $k$-centers catches up and the two remain competitive for the rest of training. Note that as the budget increases to $B = 180$ points, all of the methods converge to a final accuracy.

## E.4 DOMAIN ADAPTATION

Table 10 lists the means $\pm$ standard deviations of accuracies for all of the different source-target combinations of the Office-31 data set. We summarize the results below.

**Amazon as target data set.** For both DSLR to Amazon and Web to Amazon, our framework dominates all baselines over the entire budget range. These two domain combinations are the two hardest problems, as the unlabeled target data set, Amazon, is much larger than the source data sets and the test accuracy on Amazon is always lower than $80\%$.

**DSLR and Web as target data set.** For the other source-target combinations that have a smaller and "easier" target domain, our framework still outperforms at the extremely low budget regime $B \leq 20$ and then remains competitive with $k$-centers at the higher budgets. It is worth noting that when DSLR or Web are the target domains, the top test accuracies are near $100\%$, meaning there is little room for variation.

To further compare our framework against $k$-centers on domain adaptation, Table 11 shows the head-to-head accuracies of our model versus the baseline with $t$-tests to compare for statistical significance. The results here further validate our analysis on Table 10, as we are statistically better than $k$-centers for 15 out of a total of 42 tests and $k$-centers statistically outperforms us for only six

Table 10: Numerical values of results on domain adaptation. All entries show mean ± standard deviation over five runs. The best model for each budget range is bolded and underlined and the second best model is underlined.

**DSLR to Amazon**

| $B$ | Random | Confidence | Entropy | $k$-medoids | $k$-centers | WAAL | Wass. + EOC | Wass. + EOC + P |
|---|---|---|---|---|---|---|---|---|
| 50  | $45.9 \pm 3.2$ | $17.5 \pm 2.0$ | $17.1 \pm 2.8$ | $61.9 \pm 2.8$ | $59.7 \pm 3.0$ | $46.6 \pm 2.1$ | $\mathbf{62.5 \pm 2.4}$ | $60.0 \pm 2.8$ |
| 100 | $61.8 \pm 1.9$ | $39.0 \pm 2.3$ | $36.2 \pm 1.7$ | $66.9 \pm 2.4$ | $65.1 \pm 2.2$ | $54.6 \pm 4.2$ | $\mathbf{68.6 \pm 1.8}$ | $67.3 \pm 2.8$ |
| 150 | $68.1 \pm 0.2$ | $52.9 \pm 2.0$ | $46.9 \pm 3.6$ | $68.8 \pm 0.8$ | $68.0 \pm 1.5$ | $62.0 \pm 1.9$ | $\mathbf{71.4 \pm 1.0}$ | $71.3 \pm 1.1$ |
| 200 | $70.7 \pm 0.8$ | $60.3 \pm 2.0$ | $55.0 \pm 2.6$ | $70.2 \pm 0.9$ | $70.3 \pm 1.5$ | $66.4 \pm 2.8$ | $72.5 \pm 1.1$ | $\mathbf{73.8 \pm 0.9}$ |
| 250 | $71.2 \pm 1.0$ | $64.7 \pm 1.2$ | $62.1 \pm 4.2$ | $71.9 \pm 1.5$ | $71.4 \pm 1.7$ | $69.5 \pm 2.1$ | $73.4 \pm 0.9$ | $\mathbf{74.7 \pm 1.1}$ |
| 300 | $73.2 \pm 0.6$ | $69.0 \pm 1.3$ | $67.4 \pm 2.8$ | $72.6 \pm 1.0$ | $72.5 \pm 1.5$ | $69.8 \pm 1.4$ | $73.9 \pm 1.0$ | $\mathbf{75.5 \pm 1.3}$ |
| 350 | $73.8 \pm 1.1$ | $72.6 \pm 1.4$ | $70.4 \pm 3.7$ | $73.2 \pm 1.2$ | $73.1 \pm 1.5$ | $70.8 \pm 0.8$ | $74.7 \pm 0.8$ | $\mathbf{77.1 \pm 1.4}$ |

**Web to Amazon**

| $B$ | Random | Confidence | Entropy | $k$-medoids | $k$-centers | WAAL | Wass. + EOC | Wass. + EOC + P |
|---|---|---|---|---|---|---|---|---|
| 50  | $50.4 \pm 4.0$ | $19.2 \pm 6.7$ | $23.4 \pm 6.7$ | $64.7 \pm 2.1$ | $61.9 \pm 2.1$ | $50.4 \pm 3.3$ | $63.1 \pm 3.3$ | $\mathbf{65.9 \pm 2.3}$ |
| 100 | $61.2 \pm 3.0$ | $40.3 \pm 2.1$ | $37.5 \pm 4.7$ | $68.3 \pm 1.5$ | $67.7 \pm 2.9$ | $59.8 \pm 2.4$ | $67.8 \pm 2.4$ | $\mathbf{69.9 \pm 1.7}$ |
| 150 | $68.4 \pm 2.6$ | $50.6 \pm 3.2$ | $48.5 \pm 6.5$ | $70.4 \pm 1.4$ | $69.6 \pm 2.0$ | $65.5 \pm 1.3$ | $71.9 \pm 1.3$ | $\mathbf{72.8 \pm 1.3}$ |
| 200 | $70.9 \pm 2.0$ | $58.4 \pm 4.4$ | $58.4 \pm 3.3$ | $71.9 \pm 0.7$ | $71.1 \pm 2.1$ | $69.5 \pm 2.0$ | $73.8 \pm 2.0$ | $\mathbf{74.5 \pm 1.2}$ |
| 250 | $72.9 \pm 1.7$ | $66.9 \pm 3.4$ | $64.5 \pm 3.5$ | $72.7 \pm 0.9$ | $72.7 \pm 1.6$ | $70.2 \pm 1.8$ | $74.3 \pm 1.8$ | $\mathbf{75.7 \pm 1.5}$ |
| 300 | $74.6 \pm 1.6$ | $69.3 \pm 4.0$ | $68.4 \pm 3.7$ | $73.6 \pm 1.2$ | $73.9 \pm 1.4$ | $72.9 \pm 1.8$ | $75.7 \pm 1.8$ | $\mathbf{76.0 \pm 1.7}$ |
| 350 | $75.3 \pm 1.8$ | $73.1 \pm 2.1$ | $72.3 \pm 2.6$ | $74.3 \pm 1.3$ | $74.7 \pm 0.8$ | $73.7 \pm 1.8$ | $76.4 \pm 1.8$ | $\mathbf{77.0 \pm 1.4}$ |

**Web to DSLR**

| $B$ | Random | Confidence | Entropy | $k$-medoids | $k$-centers | WAAL | Wass. + EOC | Wass. + EOC + P |
|---|---|---|---|---|---|---|---|---|
| 10 | $31.3 \pm 5.2$ | $17.8 \pm 4.3$ | $16.4 \pm 5.4$ | $33.0 \pm 3.3$ | $26.7 \pm 0.9$ | $31.3 \pm 5.0$ | $\mathbf{36.4 \pm 4.0}$ | $30.4 \pm 1.5$ |
| 20 | $49.8 \pm 5.2$ | $31.5 \pm 3.8$ | $29.8 \pm 3.2$ | $41.4 \pm 6.8$ | $60.9 \pm 2.3$ | $42.6 \pm 5.6$ | $67.3 \pm 1.0$ | $\mathbf{67.7 \pm 3.9}$ |
| 30 | $64.3 \pm 3.8$ | $45.0 \pm 5.2$ | $43.1 \pm 5.0$ | $48.3 \pm 4.3$ | $\mathbf{96.8 \pm 1.2}$ | $60.1 \pm 6.8$ | $88.4 \pm 2.2$ | $88.3 \pm 1.4$ |
| 40 | $77.4 \pm 2.2$ | $58.8 \pm 7.1$ | $55.4 \pm 8.3$ | $54.5 \pm 1.9$ | $\mathbf{99.7 \pm 0.3}$ | $69.4 \pm 5.5$ | $95.0 \pm 0.8$ | $96.7 \pm 2.0$ |
| 50 | $81.5 \pm 2.7$ | $71.0 \pm 5.1$ | $67.0 \pm 8.6$ | $61.8 \pm 3.3$ | $\mathbf{99.6 \pm 0.5}$ | $86.0 \pm 6.2$ | $96.9 \pm 0.6$ | $99.0 \pm 1.7$ |
| 60 | $85.4 \pm 4.7$ | $81.4 \pm 2.5$ | $75.5 \pm 7.3$ | $66.2 \pm 3.4$ | $\mathbf{100 \pm 0}$ | $96.9 \pm 1.9$ | $97.7 \pm 1.9$ | $99.0 \pm 1.7$ |
| 70 | $90.0 \pm 3.4$ | $91.5 \pm 2.6$ | $85.9 \pm 4.9$ | $72.6 \pm 3.8$ | $100 \pm 0$ | $99.9 \pm 0.3$ | $97.6 \pm 2.0$ | $\mathbf{100 \pm 0}$ |

**Amazon to DSLR**

| $B$ | Random | Confidence | Entropy | $k$-medoids | $k$-centers | WAAL | Wass. + EOC | Wass. + EOC + P |
|---|---|---|---|---|---|---|---|---|
| 10 | $28.6 \pm 5.5$ | $17.5 \pm 4.9$ | $16.5 \pm 0.9$ | $30.7 \pm 4.5$ | $29.4 \pm 4.0$ | $29.4 \pm 5.3$ | $29.1 \pm 3.7$ | $\mathbf{31.1 \pm 5.9}$ |
| 20 | $45.4 \pm 4.8$ | $34.4 \pm 8.2$ | $28.2 \pm 5.4$ | $39.7 \pm 2.6$ | $54.8 \pm 4.3$ | $42.2 \pm 6.1$ | $52.6 \pm 5.1$ | $\mathbf{56.8 \pm 4.9}$ |
| 30 | $57.6 \pm 2.5$ | $48.0 \pm 7.2$ | $44.8 \pm 6.1$ | $48.2 \pm 5.6$ | $\mathbf{82.0 \pm 4.1}$ | $56.5 \pm 4.8$ | $69.3 \pm 3.2$ | $76.2 \pm 4.1$ |
| 40 | $69.8 \pm 2.5$ | $58.8 \pm 6.9$ | $55.4 \pm 4.8$ | $55.5 \pm 7.2$ | $\mathbf{87.4 \pm 3.2}$ | $70.0 \pm 3.6$ | $81.6 \pm 1.8$ | $81.9 \pm 2.8$ |
| 50 | $71.7 \pm 5.1$ | $70.6 \pm 4.0$ | $65.0 \pm 6.8$ | $58.5 \pm 8.0$ | $\mathbf{89.8 \pm 1.7}$ | $78.6 \pm 3.7$ | $87.1 \pm 3.2$ | $87.0 \pm 3.2$ |
| 60 | $77.7 \pm 4.0$ | $76.6 \pm 3.9$ | $72.6 \pm 5.9$ | $62.0 \pm 7.1$ | $\mathbf{90.6 \pm 2.7}$ | $83.6 \pm 3.3$ | $88.4 \pm 1.7$ | $89.1 \pm 3.8$ |
| 70 | $80.8 \pm 3.1$ | $83.4 \pm 2.7$ | $77.7 \pm 6.0$ | $64.8 \pm 7.2$ | $\mathbf{92.2 \pm 1.4}$ | $87.3 \pm 2.7$ | $90.3 \pm 1.3$ | $90.3 \pm 3.0$ |

**DSLR to Web**

| $B$ | Random | Confidence | Entropy | $k$-medoids | $k$-centers | WAAL | Wass. + EOC | Wass. + EOC + P |
|---|---|---|---|---|---|---|---|---|
| 10 | $28.5 \pm 2.1$ | $23.3 \pm 3.5$ | $22.2 \pm 3.2$ | $32.8 \pm 2.5$ | $30.2 \pm 2.8$ | $28.6 \pm 2.0$ | $32.4 \pm 2.6$ | $\mathbf{35.9 \pm 2.4}$ |
| 20 | $46.2 \pm 2.3$ | $32.7 \pm 5.8$ | $30.2 \pm 4.0$ | $47.8 \pm 8.0$ | $58.1 \pm 2.9$ | $45.6 \pm 6.7$ | $\mathbf{62.1 \pm 5.4}$ | $62.7 \pm 1.3$ |
| 30 | $58.0 \pm 5.7$ | $41.2 \pm 4.5$ | $35.8 \pm 5.5$ | $52.6 \pm 7.1$ | $\mathbf{93.6 \pm 1.3}$ | $56.4 \pm 3.8$ | $84.8 \pm 4.9$ | $86.6 \pm 2.8$ |
| 40 | $73.2 \pm 6.6$ | $51.5 \pm 4.5$ | $44.9 \pm 5.3$ | $57.0 \pm 5.7$ | $\mathbf{97.2 \pm 0.8}$ | $71.1 \pm 2.7$ | $93.9 \pm 2.4$ | $95.6 \pm 1.0$ |
| 50 | $78.1 \pm 5.1$ | $61.3 \pm 3.8$ | $53.7 \pm 4.2$ | $59.6 \pm 7.0$ | $\mathbf{98.0 \pm 0.9}$ | $80.2 \pm 3.4$ | $96.6 \pm 1.2$ | $96.4 \pm 1.5$ |
| 60 | $82.3 \pm 2.0$ | $69.5 \pm 6.8$ | $62.5 \pm 7.1$ | $61.6 \pm 6.8$ | $\mathbf{98.2 \pm 1.6}$ | $87.7 \pm 4.4$ | $97.8 \pm 0.8$ | $97.5 \pm 0.9$ |
| 70 | $83.5 \pm 1.5$ | $75.5 \pm 6.7$ | $71.8 \pm 5.6$ | $65.8 \pm 6.7$ | $98.6 \pm 1.1$ | $93.8 \pm 3.2$ | $\mathbf{99.2 \pm 0}$ | $97.8 \pm 1.1$ |

**Amazon to Web**

| $B$ | Random | Confidence | Entropy | $k$-medoids | $k$-centers | WAAL | Wass. + EOC | Wass. + EOC + P |
|---|---|---|---|---|---|---|---|---|
| 10 | $26.4 \pm 3.5$ | $14.9 \pm 3.2$ | $17.5 \pm 3.7$ | $31.5 \pm 4.0$ | $25.0 \pm 2.8$ | $26.6 \pm 3.1$ | $\mathbf{32.2 \pm 2.7}$ | $31.7 \pm 2.7$ |
| 20 | $41.7 \pm 1.3$ | $31.0 \pm 6.3$ | $28.0 \pm 6.5$ | $42.2 \pm 1.8$ | $53.6 \pm 3.9$ | $35.6 \pm 3.1$ | $57.6 \pm 5.7$ | $\mathbf{57.9 \pm 4.3}$ |
| 30 | $51.9 \pm 3.5$ | $43.2 \pm 7.3$ | $40.9 \pm 9.3$ | $48.4 \pm 4.3$ | $\mathbf{82.7 \pm 2.8}$ | $49.8 \pm 4.0$ | $74.1 \pm 3.9$ | $78.6 \pm 6.1$ |
| 40 | $66.5 \pm 5.7$ | $53.5 \pm 7.4$ | $51.8 \pm 8.9$ | $53.1 \pm 5.8$ | $\mathbf{87.7 \pm 3.6}$ | $59.7 \pm 4.7$ | $80.2 \pm 4.2$ | $82.6 \pm 3.2$ |
| 50 | $71.7 \pm 3.8$ | $64.2 \pm 5.7$ | $60.7 \pm 7.3$ | $58.0 \pm 7.0$ | $\mathbf{89.2 \pm 3.1}$ | $74.3 \pm 4.0$ | $84.6 \pm 1.6$ | $84.5 \pm 3.4$ |
| 60 | $75.2 \pm 1.2$ | $73.4 \pm 3.5$ | $68.9 \pm 7.9$ | $61.8 \pm 8.0$ | $\mathbf{89.7 \pm 2.6}$ | $81.4 \pm 5.3$ | $86.9 \pm 2.0$ | $84.5 \pm 2.4$ |
| 70 | $76.5 \pm 1.0$ | $81.1 \pm 1.9$ | $77.8 \pm 4.2$ | $63.2 \pm 5.8$ | $\mathbf{90.4 \pm 3.1}$ | $85.9 \pm 4.5$ | $88.5 \pm 2.3$ | $86.4 \pm 2.5$ |

tests. More specifically, we outperform $k$-centers on five out of the seven budgets for both DSLR to Amazon and Web to Amazon, which is the hardest domain to learn, especially since the source data sets are much smaller than the target. Note that the remaining source-target combinations are easy problems that nominally yield high accuracies with unsupervised domain adaptation, meaning that

Table 11: Head-to-head comparison of Wass. + EOC + P versus $k$-centers on domain adaptation with statistical $t$-tests to compare. The best model for each budget range is bolded and underlined. Tests where Wass. + EOC + P statistically outperforms $k$-centers are marked in green and tests where $k$-centers statistically outperforms Wass. + EOC + P are marked in red.

| DSLR to Amazon | | | | | | | |
|---|---|---|---|---|---|---|---|
| $B$ | 50 | 100 | 150 | 200 | 250 | 300 | 350 |
| $k$-centers | $59.7 \pm 3.0$ | $65.1 \pm 2.2$ | $68.0 \pm 1.5$ | $70.3 \pm 1.5$ | $71.4 \pm 1.7$ | $72.5 \pm 1.5$ | $73.1 \pm 1.5$ |
| Wass. + EOC + P | $\mathbf{60.0 \pm 2.8}$ | $\mathbf{67.3 \pm 2.8}$ | $\underline{\mathbf{71.3 \pm 1.1}}$ | $\underline{\mathbf{73.8 \pm 0.9}}$ | $\underline{\mathbf{74.7 \pm 1.1}}$ | $\underline{\mathbf{75.5 \pm 1.3}}$ | $\underline{\mathbf{77.1 \pm 1.4}}$ |
| $p$-value $< 0.05$ | | | ✓ | ✓ | ✓ | ✓ | ✓ |

| Web to Amazon | | | | | | | |
|---|---|---|---|---|---|---|---|
| $B$ | 50 | 100 | 150 | 200 | 250 | 300 | 350 |
| $k$-centers | $61.9 \pm 2.1$ | $67.7 \pm 2.9$ | $69.6 \pm 2.0$ | $71.1 \pm 2.1$ | $72.7 \pm 1.6$ | $73.9 \pm 1.4$ | $74.7 \pm 0.8$ |
| Wass. + EOC + P | $\underline{\mathbf{65.9 \pm 2.3}}$ | $\underline{\mathbf{69.9 \pm 1.7}}$ | $\underline{\mathbf{72.8 \pm 1.3}}$ | $\underline{\mathbf{74.5 \pm 1.2}}$ | $\underline{\mathbf{75.7 \pm 1.5}}$ | $\underline{\mathbf{76.0 \pm 1.7}}$ | $\underline{\mathbf{77.0 \pm 1.4}}$ |
| $p$-value $< 0.05$ | ✓ | | ✓ | ✓ | ✓ | | ✓ |

| Web to DSLR | | | | | | | |
|---|---|---|---|---|---|---|---|
| $B$ | 10 | 20 | 30 | 40 | 50 | 60 | 70 |
| $k$-centers | $26.7 \pm 0.9$ | $60.9 \pm 2.3$ | $\underline{\mathbf{96.8 \pm 1.2}}$ | $\underline{\mathbf{99.7 \pm 0.3}}$ | $\underline{\mathbf{99.6 \pm 0.5}}$ | $\underline{\mathbf{100 \pm 0}}$ | $\underline{\mathbf{100 \pm 0}}$ |
| Wass. + EOC + P | $\underline{\mathbf{30.4 \pm 1.5}}$ | $\underline{\mathbf{67.7 \pm 3.9}}$ | $88.3 \pm 1.4$ | $96.7 \pm 2.0$ | $99.0 \pm 1.7$ | $99.0 \pm 1.7$ | $\underline{\mathbf{100 \pm 0}}$ |
| $p$-value $< 0.05$ | ✓ | ✓ | ✗ | ✗ | | | |

| Amazon to DSLR | | | | | | | |
|---|---|---|---|---|---|---|---|
| $B$ | 10 | 20 | 30 | 40 | 50 | 60 | 70 |
| $k$-centers | $29.4 \pm 4.0$ | $54.8 \pm 4.3$ | $\underline{\mathbf{82.0 \pm 4.1}}$ | $\underline{\mathbf{87.4 \pm 3.2}}$ | $\underline{\mathbf{89.8 \pm 1.7}}$ | $\underline{\mathbf{90.6 \pm 2.7}}$ | $\underline{\mathbf{92.2 \pm 1.4}}$ |
| Wass. + EOC + P | $\underline{\mathbf{31.1 \pm 5.9}}$ | $\underline{\mathbf{56.8 \pm 4.9}}$ | $76.2 \pm 4.1$ | $81.9 \pm 2.8$ | $87.0 \pm 3.2$ | $89.1 \pm 3.8$ | $90.3 \pm 3.0$ |
| $p$-value $< 0.05$ | | | | ✗ | | | |

| DSLR to Web | | | | | | | |
|---|---|---|---|---|---|---|---|
| $B$ | 10 | 20 | 30 | 40 | 50 | 60 | 70 |
| $k$-centers | $30.2 \pm 2.8$ | $58.1 \pm 2.9$ | $\underline{\mathbf{93.6 \pm 1.3}}$ | $\underline{\mathbf{97.2 \pm 0.8}}$ | $\underline{\mathbf{98.0 \pm 0.9}}$ | $\underline{\mathbf{98.2 \pm 1.6}}$ | $\underline{\mathbf{98.6 \pm 1.1}}$ |
| Wass. + EOC + P | $\underline{\mathbf{35.9 \pm 2.4}}$ | $\underline{\mathbf{62.7 \pm 1.3}}$ | $86.6 \pm 2.8$ | $95.6 \pm 1.0$ | $96.4 \pm 1.5$ | $97.5 \pm 0.9$ | $97.8 \pm 1.1$ |
| $p$-value $< 0.05$ | ✓ | ✓ | ✗ | ✗ | | | |

| Amazon to Web | | | | | | | |
|---|---|---|---|---|---|---|---|
| $B$ | 10 | 20 | 30 | 40 | 50 | 60 | 70 |
| $k$-centers | $25.0 \pm 2.8$ | $53.6 \pm 3.9$ | $\underline{\mathbf{82.7 \pm 2.8}}$ | $\underline{\mathbf{87.7 \pm 3.6}}$ | $\underline{\mathbf{89.2 \pm 3.1}}$ | $\underline{\mathbf{89.7 \pm 2.6}}$ | $\underline{\mathbf{90.4 \pm 3.1}}$ |
| Wass. + EOC + P | $\underline{\mathbf{31.7 \pm 2.7}}$ | $\underline{\mathbf{57.9 \pm 4.3}}$ | $78.6 \pm 6.1$ | $82.6 \pm 3.2$ | $84.5 \pm 3.4$ | $84.5 \pm 2.4$ | $87.4 \pm 2.5$ |
| $p$-value $< 0.05$ | ✓ | | | | | ✗ | |

the marginal value of labeling points is much less for these tasks. For Web to DSLR and DSLR to Web, we are statistically better at smaller budgets whereas $k$-centers is better for medium budgets.

### E.5 CLASSICAL ACTIVE LEARNING

Here, we ablate the effect of feature generation by removing unsupervised pre-training and considering classical batch-mode active learning (Sener & Savarese, 2017; Sinha et al., 2019; Shui et al., 2020). Our baseline, WAAL, is an alternative active learning strategy that also minimizes the Wasserstein distance but rather by using an adversarial model to estimate the dual distance (Shui et al., 2020). WAAL is designed specifically for classical active learning where self-supervised features are not available. In this experiment, we follow their exact setup to demonstrate (i) that optimization is effective over a greedy approach such as WAAL even in the high-budget setting, and (ii) that optimization is still effective when given weaker features via supervised learning as opposed to powerful self-supervised features.

**Methods.** We initialize our labeled data set with a small randomly selected subset. Then in each round, we first train the classifier using supervised learning on the currently labeled pool and then use the final layer of the trained classifier to obtain feature embeddings. These embeddings are then used to solve our optimization problem and select the next $B$ points to label. We use the available code of Shui et al. (2020) to implement WAAL without changing any of their parameters. We use their recommended model, training, and algorithm hyper-parameter settings.

Table 12: Runtime in minutes:seconds for all of the baselines on CIFAR-10. Times are averaged over five runs.

| $B$ | Random | Confidence | Entropy | $k$-medoids | $k$-centers | WAAL |
|-----|--------|-----------|---------|-------------|-------------|------|
| 10  | 0.03 | 0.03 | 0.03 | $7:56.81$  | 0.09 | 0.03  |
| 20  | 0.03 | 0.02 | 0.01 | $15:22.28$ | 0.15 | 11.43 |
| 40  | 0.02 | 0.01 | 0.01 | $7:25.69$  | 1.33 | 12.18 |
| 60  | 0.04 | 0.02 | 0.02 | $2:18.51$  | 0.34 | 11.93 |
| 80  | 0.03 | 0.02 | 0.02 | $2:18.37$  | 1.75 | 11.97 |
| 100 | 0.03 | 0.02 | 0.02 | $2:21.84$  | 0.54 | 12.25 |
| 120 | 0.03 | 0.02 | 0.01 | $2:17.18$  | 1.31 | 11.69 |
| 140 | 0.03 | 0.02 | 0.02 | $2:20.91$  | 1.19 | 11.73 |
| 160 | 0.03 | 0.02 | 0.01 | $2:18.31$  | 3.50 | 11.59 |
| 180 | 0.03 | 0.03 | 0.02 | $2:13.74$  | 0.74 | 11.51 |

In order to adapt our optimization problem to the high-budget setting, we make one modification to our algorithm. Rather than solving for the entire labeling budget in one problem, we instead randomly partition the data set into two subsets and solve two optimization problems each for half of the labeling budget respectively. For example, if we have a previously labeled set of $2,000$ images, an unlabeled pool of $48,000$, and a budget of $B = 1,000$ we randomly split the problem into two sets of $1,000$ labeled images, $24,000$ unlabeled images, and $B = 500$ budgets. This ensures that our optimization problem remains manageable in computational size. Furthermore, we implement Wass. + EOC + P, set $(\beta^+, \beta^-) = (0.6, 0.9)$, and a total time limit of $T = 4$ hours for each problem.

**Results.** From Table 4, for both data sets and at all budgets, our approach outperforms the baseline by up to $3\%$. Recall that WAAL (i) learns a feature embedding that estimates the Wasserstein distance via dual regularization and (ii) uses a greedy uncertainty strategy to select points that minimize the Wasserstein distance of selected points. In contrast, we use standard embeddings and formulate an integer program that to select points that minimize the Wasserstein distance. Our approach can provably obtain optimal solutions. Our performance highlights the value of optimization via improvement in classification accuracy. Furthermore, we conclude that our optimization problem can be tractably solved in higher budget settings.

### E.6   WALL-CLOCK ANALYSIS

Table 12 presents the average runtime for each active learning baseline. The runtimes do not include the first cost of generating features, which occurs for each method except for Random. Nearly every baseline requires only a few seconds to make queries. The $k$-medoids baseline usually requires several minutes due to the cost of computing the distance matrices between points. In contrast, we encourage running our method for at least 20 minutes and up to several hours.

A potential concern for the use of our method in active learning is the long run time (i.e., requiring several hours) in contrast to baselines that typically require on the order of seconds. In general, combinatorial optimization frameworks for deep learning usually scale poorly with data set size. For example, the nominal MILP (4) does not even fit into memory for large data sets such as CIFAR-10 and SVHN. GBD simplifies our problem by permitting a variable runtime and easier computational burdern, while still preserving global optimality. Ultimately, we emphasize that the time cost of selection can be less than model training time and in some applications, less than human labeling time itself (Rajchl et al., 2016). Furthermore, the selection is entirely automated and offline from training. Thus, we suggest that it is more practical to ensure the best possible selection regardless of runtime in order to minimize the real human labor of labeling.

### E.7   ADDITIONAL ABLATIONS

**Customizing the Wasserstein distance.** Figure 6 plots accuracy on CIFAR-10 using Wass. + EOC with Cosine versus Euclidean distance. The Euclidean alternative outperforms Cosine distance only when $B = 10$ whereas Cosine distance outperforms the alternative for higher budgets. Using Cosine distance yields better downstream accuracy here because the representation learning step, SimCLR, also maximizes a Cosine distance between different images.

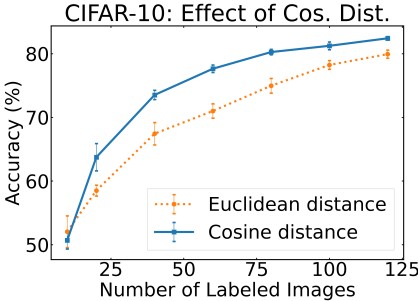

Figure 6: Ablation of using Cosine distance as the base metric in the Wasserstein distance problem for CIFAR-10.

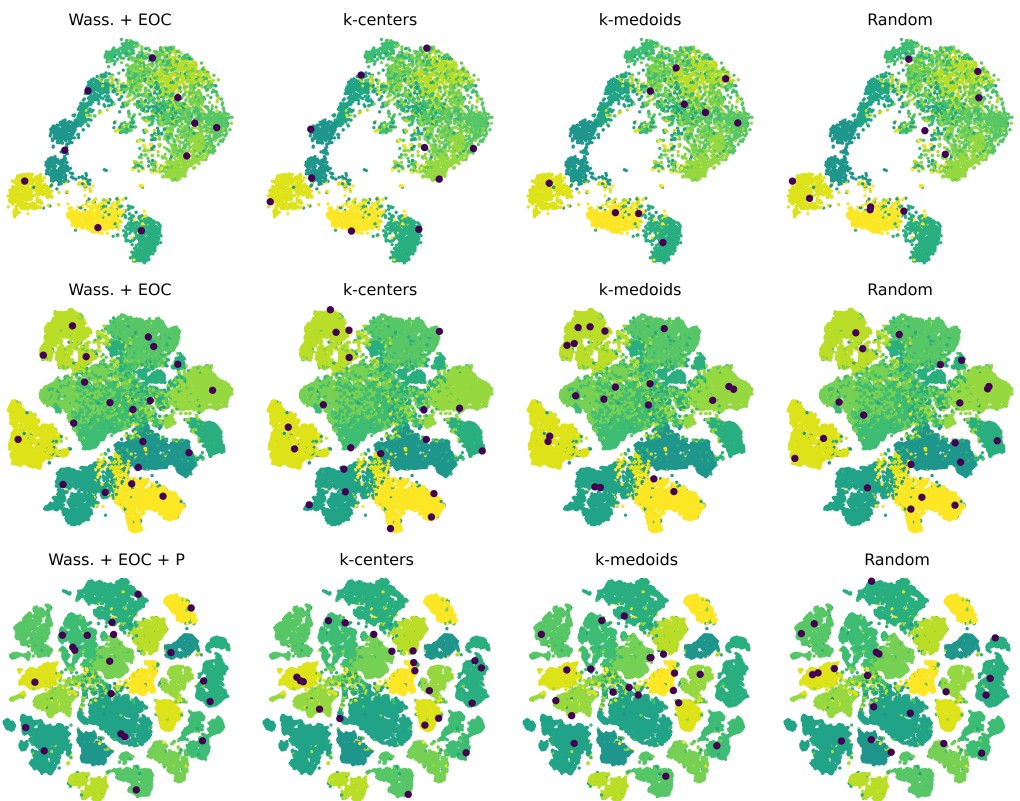

Figure 7: $t$-SNE visualizations of the latent space obtained from pre-training. Images selected by each strategy are marked. The first row shows STL-10 at $B = 10$, the second row shows CIFAR-10 at $B = 20$, and the third row shows SVHN at $B = 20$.

**Qualitative Analysis.** Figure 7 shows $t$-SNE visualizations of the latent space in the first and second round for STL-10, CIFAR-10, and SVHN, respectively. In each case, our selection strategy provides a better coverage of the entire latent space by selecting points closer to cluster centers and covering most clusters. All of the baselines leave large holes in certain regions of the latent space.

Figures 8, 9, and 10 show a sample set of labeled images by the final round for each data set. In each case, our model covers every class with a labeled image within the first two rounds. For example with STL-10, the first round labels two images of dogs instead of a cat, and so the second round corrects for this issue by identifying and labeling two cats. We observe the same trend with airplanes and cars for CIFAR-10. Furthermore, we select images from nearly every class in each round.

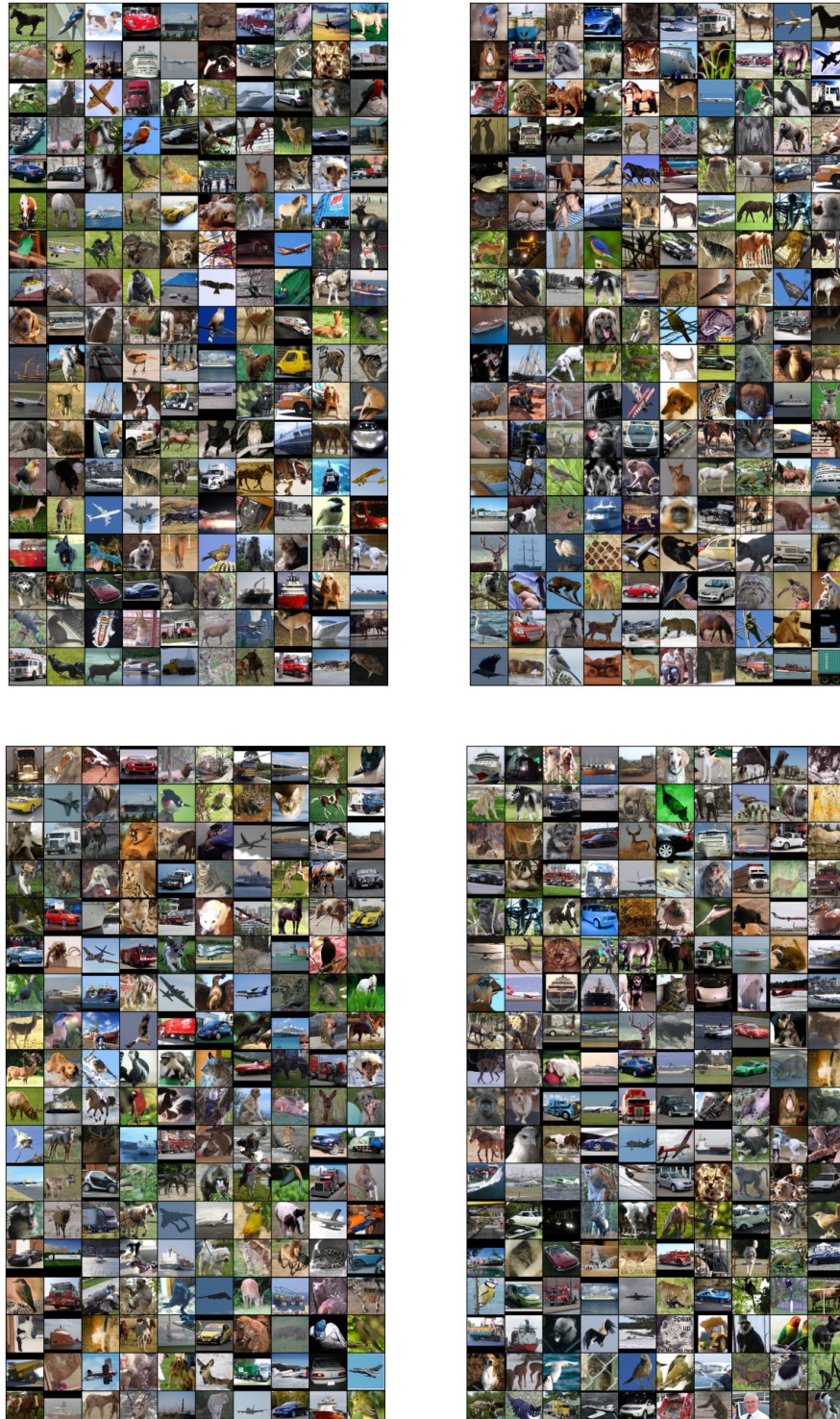

Figure 8: Images selected for labeling on STL-10 with different methods: Wass. + EOC (top left), $k$-centers (top right), $k$-medoids (bottom left), Random (bottom right). The first two rows were selected in the first two rounds and every two rows after were selected in the subsequent rounds.

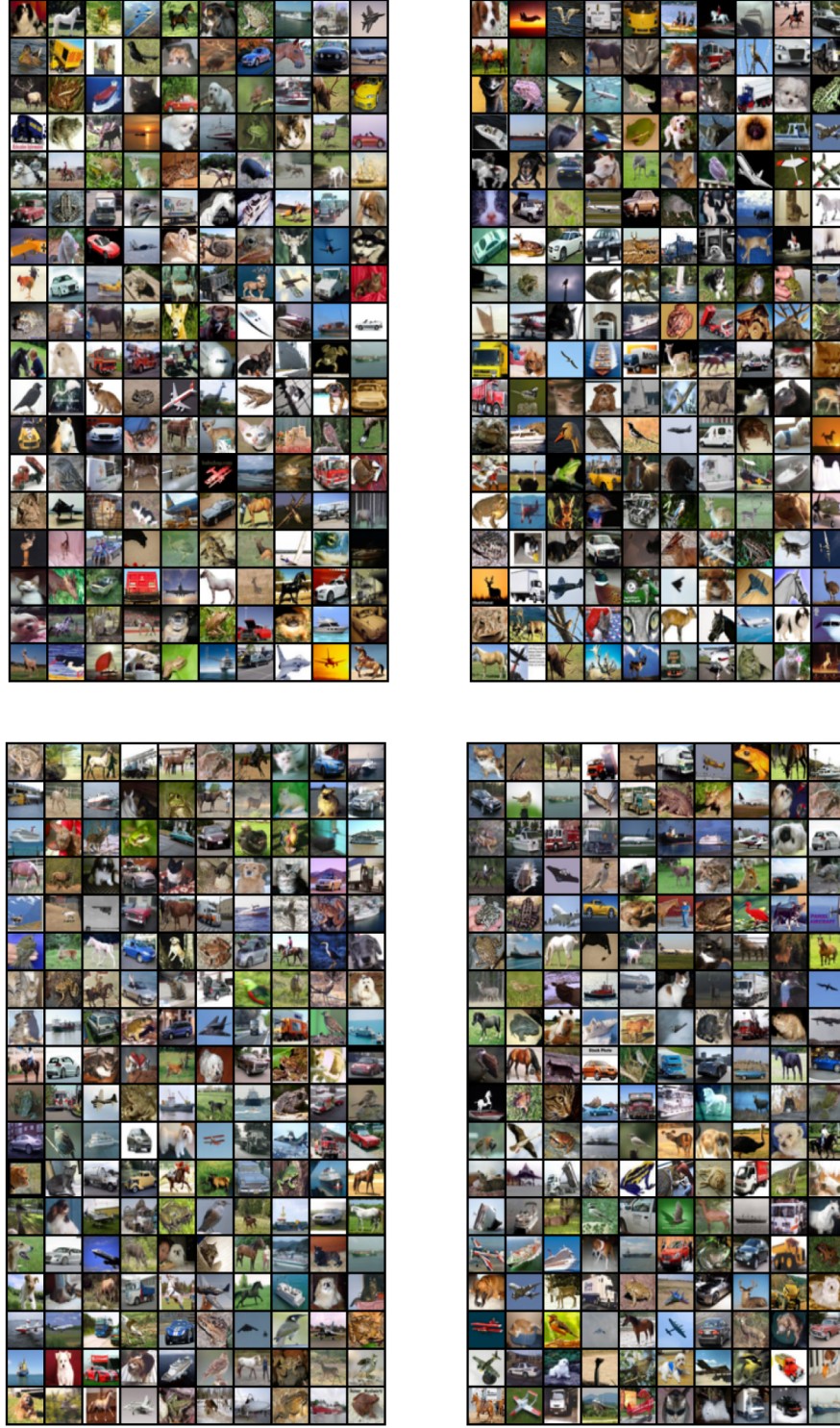

Figure 9: Images selected for labeling on CIFAR-10 with different methods: Wass. + EOC (top left), $k$-centers (top right), $k$-medoids (bottom left), Random (bottom right). The first two rows were selected in the first two rounds and every two rows after were selected in the subsequent rounds.

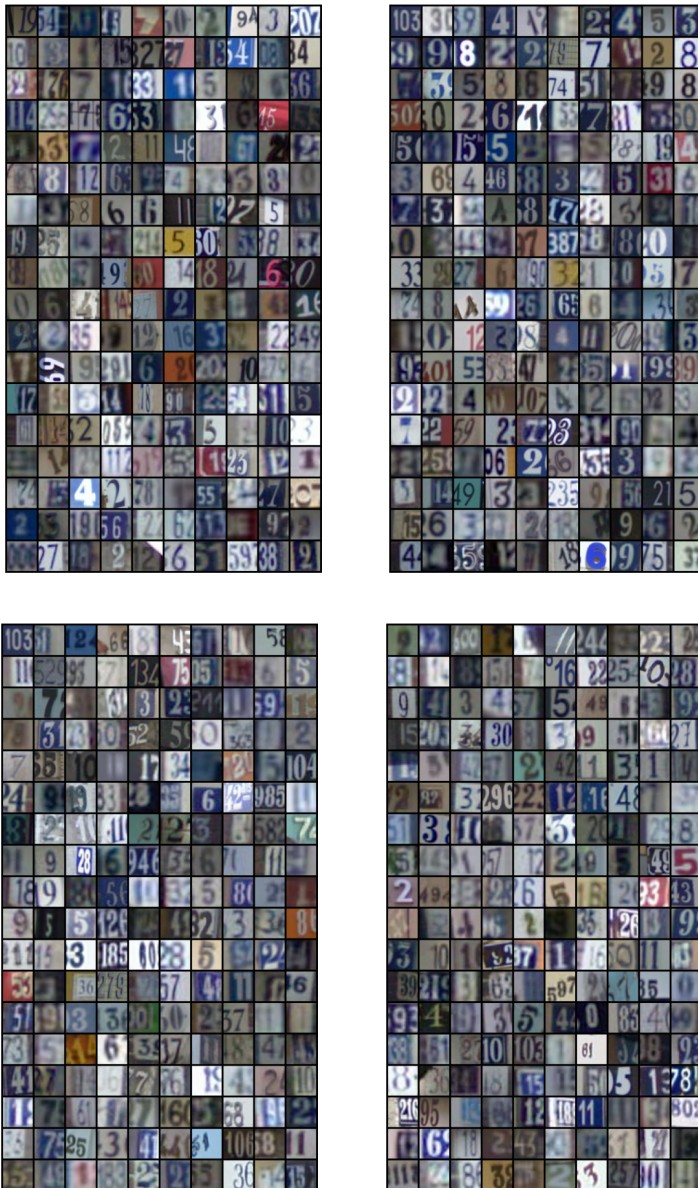

Figure 10: Images selected for labeling on SVHN with different methods: Wass. + EOC + P (top left), $k$-centers (top right), $k$-medoids (bottom left), Random (bottom right). The first two rows were selected in the first two rounds and every two rows after were selected in the subsequent rounds.

Figure 8 compares our method versus $k$-centers, $k$-medoids, and Random for STL-10. Compared to the baselines, we identify and label examples from more classes in the first rounds. While we capture labeled examples of all classes by the second round, the $k$-centers baseline does not label a dog until the third round. Note that in the first round, the Random baseline includes three images of dogs, meaning that this baseline at small budgets cannot accurately select a diverse set of points. In particular in the first rounds, our approach selects unobstructed images that capture the silhouette of each class. For example in the first round we label unobostructed full examples of horse, bird, dog, car, deer, and airplane, whereas the $k$-centers baseline includes unobstructed bird, deer, airplane, and horse, the $k$-medoids baseline includes only an unobstructed dog and car, and the Random baseline includes an unobstructed ship, dog, and truck. Finally, the pictures selected by all three non-random strategies appear brighter and sharper on average than the Random baseline.

Figure 9 compares our method versus the three baselines for CIFAR-10. The samples here display similar trends to the STL-10 example: (i) we more quickly label a diverse set of classes, (ii) our approach identifies unobstructed, clear images in the first rounds, and (iii) the non-random alternatives select brighter images more often. Specifically, we cover all classes within the second round; in contrast, the $k$-medoids baseline requires three rounds to cover all classes. Furthermore in the first round, we obtain unobstructed images of a dog, horse, deer, frog, truck, and airplane, while the $k$-centers baseline obtains an unobstructed airplane, bird, truck, and horse, and the $k$-medoids baseline obtains an unobstructed frog, horse, truck.

Finally, Figure 10 compares our method versus the three baselines for SVHN. Here in the first round, our model more often selects different classes (i.e., 0, 1, 2, 3, 4, 5, 7, 9) versus the baselines. Furthermore, in the early rounds, our images cover a variety of different backgrounds including blue, red, and yellow.

