# OpenReview forum: "Low-Budget Active Learning via Wasserstein Distance: An Integer Programming Approach"
_ICLR.cc/2022/Conference — ICLR 2022 Poster_

### Official Review · Reviewer_HRFS · 2021-11-01

**Correctness:** 4
**Technical Novelty And Significance:** 2
**Empirical Novelty And Significance:** 2
**Recommendation:** 6
**Confidence:** 4

**Main Review:**

The paper is well-written and easy to follow. The authors provide an integer programming based approach for representation-based active learning, which shows empirical advantages over existing ones. I summarize my questions as follows.

1. In section 3.2, the authors compare the size of the original optimization problem in Eq (4) and the relaxed one. In the relaxed problem, however, one needs to compute the Wasserstein distance at each iteration. What is the computational complexity of computing the mentioned Wasserstein distance? It would be great to include the explicit complexity, rather than just saying that ``efficient algorithms exist''.
2. The author mentioned that the proposed method usually takes longer to make selections. I wonder how well the proposed method performs when it is given the same running time as other baselines, e.g., 3 minutes for both k-medoids and the proposed method?


**Summary Of The Paper:**

This paper studies active learning from a mixed integer programming perspective. The active learning strategy is representation-based, and aims at selecting a core set that minimizes the Wasserstein distance. Various tricks, e.g., enhanced optimality cuts and pruning, from the integer programming literature are used to accelerate the algorithm. Empirical results show advantages of the proposed algorithm over existing ones.

**Summary Of The Review:**

I think overall the approach makes sense, but I didn't find significantly novel contributions, either theoretically or empirically (maybe the authors could point out their novel contributions). As a result, I'll vote for a weak acceptance.

---

> ### Author Response · Authors · 2021-11-19
> **Thank you for the review**
>
> Thank you for the positive feedback. We have updated the paper with revisions (in red).
>
> **Computational cost:** In our revision, we include a new Appendix D.3 that discusses the complexity of the algorithm. Computing Wasserstein distances comprises of two components:
> -	A one-time cost of computing the base distance matrix, i.e., $O(dN^2)$ where $d$ is the feature dimension. We do this at the start of the integer programming algorithm and then store this matrix in memory.
> -	Computing the Wasserstein distance LP in each iteration. This is typically solved via the Network Simplex Algorithm $O(N^3(\log N)^2)$ (Peyre and Cuturi 2019).
>
> In our experiments, we use the off-the-shelf solver, POT: Python Optimal Transport library (Flamary et al. 2021). Computing Wasserstein distances in our experiments requires up to 30 seconds at most. Furthermore we restrict the W-RMP solver to 180 seconds, so a single iteration of the algorithm requires between 40-210 seconds (see Figure 4 (right) for a detailed ablation study on the W-RMP solver time out). In contrast, the original problem (4) does not even fit into our memory when $N=50,000$.
>
> **Further restricting the runtime:** Our revision includes results on CIFAR-10 as we restrict the runtime down to 3 minutes (summarized below). We recall that k-medoids is a heuristic to solve problem (4) and show that even with a 3 minute restriction, we find better solutions (i.e., with lower objective function value) and better downstream classifiers than k-medoids. Nonetheless, from Figure 4 (right), we typically suggest running the algorithm for at least 20 minutes which will allow enough iterations to attain a high-quality selection.
>
> Table 1. Accuracy on CIFAR-10 (c.f. Figure 4 in the paper)
>
> | B  | 3 min. | 6 min. | 3 hr. | k-medoids |
> |-----|----------|----------|-------|---------------|
> | $10$  | $42.7 \pm 4.2$ | $37.6 \pm 3.4$ | $50.7 \pm 3.0$ | $45.6 \pm 5.9$ |
> | $20$  | $56.6 \pm 1.4$ | $57.3 \pm 1.9$ | $63.7 \pm 4.8$ | $53.7 \pm 7.0$ |
> | $40$  | $68.6 \pm 7.9$ | $70.3 \pm 3.7$ | $73.5 \pm 1.6$ | $65.3 \pm 3.0$ |
> | $60$  | $71.2 \pm 6.1$ | $73.7 \pm 4.2$ | $77.6 \pm 1.3$ | $69.4 \pm 3.2$ |
> | $80$  | $74.4 \pm 3.2$ | $77.2 \pm 2.5$ | $80.3 \pm 0.9$ | $71.0 \pm 1.5$ |
> | $100$ | $78.7 \pm 1.2$ | $79.5 \pm 0.8$ | $81.2 \pm 1.4$ | $72.6 \pm 0.9$ |
> | $120$ | $80.9 \pm 1.3$ | $81.5 \pm 0.7$ | $82.4 \pm 0.6$ | $74.0 \pm 0.8$ |
>
> Table 2. Objective function value (Wasserstein distance) on CIFAR-10 (c.f. Table 2 in the paper)
>
>
> | B  | 3 min. | 6 min. | 3 hr. | k-medoids |
> |-----|----------|----------|-------|---------------|
> | $120$ | $0.082$ | $0.072$ | $0.064$ | $0.095$ |
> | $140$ | $0.073$ | $0.068$ | $0.059$ | $0.089$ |
> | $160$ | $0.084$ | $0.079$ | $0.072$ | $0.099$ |
> | $180$ | $0.08$ | $0.075$ | $0.072$ | $0.096$ |
>
> **Novel contributions:**  This work includes two methodological contributions. First, we develop a new bound on the core set loss as a function of the Wasserstein distance. While previous bounds exist, ours is particularly simple (i.e., deterministic, only requires a standard Lipschitz assumption, and leverages only LP duality). Second, we develop a convergent integer programming algorithm for minimizing this Wasserstein distance. To the best of our knowledge, solving the minimum (exact) Wasserstein distance has previously only been approached with heuristics or approximations for problems of our size. Finally, our empirical contributions demonstrate the value of using optimization for active learning problems, especially in low-budget scenarios where the marginal sub-optimality of poor selections is high.
>
>
> **References**
>
> Peyre and Cuturi, Computational Optimal Transport, Foundations and Trends in Machine Learning, 2019
>
> Flamary et al, POT Python Optimal Transport library, JMLR, 2021, URL:https://pythonot.github.io/

---

### Official Review · Reviewer_9PdP · 2021-11-03

**Correctness:** 3
**Technical Novelty And Significance:** 3
**Empirical Novelty And Significance:** 2
**Recommendation:** 6
**Confidence:** 3

**Main Review:**

**Strong Points**

- The paper discusses why Wasserstein distance is minimized (Theorem 1).
- The experiments show the effectiveness of the proposed method in downstream task (active learning).
- Presentation is clear.

**Weak Points**

- [Effect of the embedding method is not investigated] SimCLR is used for obtaining the encoding but I am curious what happens if other embedding methods are used. Intuitively, the embedding is very important because the proposed method assumes that nearby points in the embedding space have the same label. In the second paragraph of Section 5, the authors say "our approach is also effective without self-supervised features"; what is the meaning of this sentence?
- [Scalability] Scalability of the proposed method seems to be low. The experiments include relatively small datasets. Does the proposed method scale even for datasets including millions of data points or larger budget values? For example, Table includes larger B's results (from 1000 to 6000); how large is the computation cost? This concern is mainly from real application scenarios; ML projects usually have budget to label more than 1000 data points (and much larger unlabeled data pool for which we need to compute Wasserstein distance).

**Questions**

- The inequality in pp.2 seems to be equal. In what sense do you argue that the RHS upper-bounds the generalization error?
- How do we solve W-RMP? Do we use ILP solver?
- This paper deals with the batch mode active learning. Is it possible to extend the proposed method to the classical sequential active learning?

**Summary Of The Paper:**

This paper proposes a (batch mode) active learning method that chooses a subset of data points to be labeled through approximating the whole dataset in terms of Wasserstein distance, which is an upper-bound of the generalization error. The selection is formulated as a large-scale mixed integer programming, and the authors propose to solve it by the GBD. For acceleration, some additional constraints are proposed. Experimental results show that the proposed method is better (or competitive) than baseline methods like k-center, k-medoids, and WAAL.

**Summary Of The Review:**

This paper is well-written and motivated, with a theoretical guarantee.
Before seeing the other reviews, I am positive to accept the paper although including some concerns (e.g., scalability).
Please answer and clarify my concerns described above.

---

> ### Author Response · Authors · 2021-11-19
> **Thank you for the review**
>
> Thank you for the positive feedback. We have updated the paper with revisions (in red).
>
> **Effect of the embedding method (SimCLR):** Due to its popularity, we used SimCLR to pretrain our representations in the standard classification task. However, our ablations show that our approach is not dependent on SimCLR and can be used with other embedding methods for pretraining (see “Domain Adaptation” paragraph in page 8 and “Representations from classical active learning” paragraph in page 9).
> -	**Domain Adaptation:** we obtain embeddings from f-DAL (Acuna et al. 2021), which is a domain adaptive representation learning framework exploiting a labeled source dataset and a (different but related) unlabeled target dataset. Even with this different embedding, we find the same trends as before in that our best approach is either the best or second best for almost all budgets (see Figure 3 for details).
> -	**Classical Active Learning:** we use the standard approach of obtaining features from the model in the previous round of supervised active learning. Our method here slightly outperforms WAAL (see Table 4), which was designed for the classical setting.
>
> We have clarified the statement in Section 5 to emphasize that we mean our approach also works with different embedding methods outside of SimCLR.
>
> **Scalability:** As observed by the reviewer, our general approach scales to larger budgets, although with increased computation. For instance with the results in Table 4, we can partition the larger problem into two smaller sub-problems (e.g., rather than selecting 5,000 points out of 50,000, instead partition to two sets of 25,000 and select 2,500 from each). Altogether, we employ a total 8 hours per active learning round (see Appendix E.5 for details).
>
> In our revision, we include a new Appendix D.3 that discusses the complexity of the algorithm. With larger data sets of 1M images, we find computing Wasserstein distances to be the primary bottleneck. This comprises of:
> -	A one-time cost of computing the base distance matrix, i.e., $O(dN^2)$ where $d$ is the feature dimension. We do this at the start of the integer programming algorithm and then store this matrix in memory.
> -	Computing the Wasserstein distance LP in each iteration. This is typically solved via the Network Simplex Algorithm, i.e., $O(N^3(\log N)^2))$ (Peyre and Cuturi 2019).
>
> As efficiently computing large-scale Wasserstein distances or their approximations is an active research area (e.g., Altschuler et al. 2019, Xie et al. 2019), we hope to improve our algorithm with new acceleration tricks in future work.
>
> We also emphasize that selecting which points to label is entirely automated, performed offline from training, and can require less time than actual training. Thus, in settings where labeling itself is prohibitively expensive (e.g., medical imaging) or where data collection is naturally difficult (e.g., domain adaptation), it would be prudent to ensure automated querying identifies the best points to label regardless of runtime.
>
> **Inequality in pp.2:** Thank you for identifying this point. We can write the current equation as an equality. In the original paper of Sener and Saverese (2018), it was written as an inequality after applying absolute values over the generalization error and core set loss. We have revised the bound to ensure it is an inequality and is more in line with the prior work.
>
> **Solving W-RMP:** Yes, we solve W-RMP as a MILP. For any fixed $\lambda$, the right-hand-side $\inf$ in the constraint of W-RMP can be solved independently of $\pi$ (due to the linearity of the inner problem), meaning that the constraints of W-RMP are always just linear constraints.
>
> **Extending to sequential active learning:** The classical sequential active learning process is the special case where one sample is annotated at a time (i.e., the difference of budgets between two rounds is 1). Our paper considers cases where this difference of budgets is 10, 20, 500, 1000 but 1 is also doable. The optimization problem and solution algorithm will still hold and likely be easy to solve, since the number of feasible solutions is combinatorial in $B$. Nonetheless, practical sequential active learning may require shorter selection runtimes than our approach provides.
>
> **References**
>
> Acuna et al., f-Domain-Adversarial Learning, ICML 2021
>
> Peyre and Cuturi, Computational Optimal Transport, Foundations and Trends in Machine Learning, 2019
>
> Altschuler et al., Massively Scalable Sinkhorn Distances via the Nystrom Method, NeurIPS 2019
>
> Xie et al., A fast proximal point method for computing exact Wasserstein distance, UAI 2019
>
> Sener and Saverese, Active Learning for Convolutional Neural Networks: A Core-Set Approach, ICLR 2018

---

### Official Review · Reviewer_3bUG · 2021-11-05

**Correctness:** 4
**Technical Novelty And Significance:** 3
**Empirical Novelty And Significance:** 3
**Recommendation:** 8
**Confidence:** 3

**Main Review:**

Strength

The paper is well written. It describes the problem in detail. I believe using Wasserstein distance to bound the core set loss is a unique and interesting way of dealing with the active learning problem. The approaches are also backed by strong empirical analysis. I find this a very strong paper. I did not get to check all the proofs but I have checked some of them and they seem correct.

Weaknesses

I dont find any weaknesses of the paper. This seems very relavant work and a strong accept in my view.

**Summary Of The Paper:**

This work tackles the problem of active learning. In every iteration the subset of the unlabeled data pool that needs to be labeled is selected by posing it as an Integer Programming (IP) problem that minimizes the discrete Wasserstein distance. Generalized Benders Decomposition algorithm is used to solve this IP through relaxations. It is shown to converge and also some acceleration techniques are provided. All the above are supported through extensive empirical analysis.

**Summary Of The Review:**

Overall a strong accept for tackling an important problem.

---

> ### Author Response · Authors · 2021-11-19
> **Thank you for the review**
>
> Thank you for the positive feedback. We have updated the paper with revisions (in red). We are happy to answer any further questions if they come up.

---

### Official Review · Reviewer_vpVq · 2021-11-05

**Correctness:** 3
**Technical Novelty And Significance:** 3
**Empirical Novelty And Significance:** 3
**Recommendation:** 6
**Confidence:** 3

**Main Review:**

Strengths:
-- Tackles active learning from a slightly different approach.
-- Wasserstein distance well justified and bounds of the integer optimisation problems are well specified (proof not checked carefully however)
-- Reasonable improvements over previous approaches:
-- Well written paper: clear, algorithms and pseudo-code provided for ease of understanding and reproducibility.
-- Thorough evaluation in low data regime against other SOTA active learning methods, in a number of scenarios (with SSL, classical active learning setting and domain adaptation).
-- Time and complexity analysis provided

Weaknesses/Potential Areas of Improvement
-- Stated that there is improvement for high budget settings, but no evidence is provided in the text of that?
-- In fact, it seems like a trend for a drop-off in performance in terms of accuracy  as budget increases; would be nice to see that even it does occur.
-- There is limited analysis of scaling to larger unlabelled datasets or datasets with more classes. Skeptical that performance would be good, or that the method would scale well. Although, again training time is often cheaper than labelling so this is not necessarily a problem, but some experiment here would be useful.

Other questions:
-- Were the features used for greedy k-centers the same as the features used in your approach? Or did you continue to use the VGG16 features that was used in Sener and Savarese?

**Summary Of The Paper:**

The paper frames active learning as an integer optimization problem, that minimises the distance Wasserstein distance between unlabelled pool of data. This is done in a feature space (in this case trained using self-supervised methods all the data). The method outperforms existing active learning methods for very small labelling budgets, with theoretical guarantees of integer optimisation problem (although not the performance in terms of model accuracy itself).

**Summary Of The Review:**

Well written and polished paper with novel angle on the active learning problem, with some theoretical guarantees and good experimental performance in low budget scenarios. Would like see some additional results, but this a good paper.

---

> ### Author Response · Authors · 2021-11-19
> **Thank you for the review**
>
> Thank you for the positive feedback. We have updated the paper with revisions (in red).
>
> **Improvement of our approach for high budget settings compared to baselines:** We agree that the highest relative gains for our approach versus baselines are obtained when the budgets are the smallest. In this regime, each selected image represents a large fraction of the budget, meaning here, optimally selecting which points to label will have the most impact.
> In Table 4, we show a noticeable improvement in the Classical Active Learning experiments for larger budgets (e.g., for a large budget of $B = 5,000$) over WAAL, which is a method designed specifically for the classical setting. This convinces us that our approach has a place even at larger budgets, especially if the cost of labeling each image itself is expensive.
>
> **Scaling to larger unlabeled data sets:** We agree that scalability to larger data sets (e.g., 1M images) presents a bottleneck, specifically because solving Wasserstein distances requires computing and storing large distance matrices. This is an active research area and as more efficient Wasserstein computation methods (or their approximations) arrive (e.g., Altschuler et al. 2019, Xie et al. 2019), we hope to make our approach more scalable. For now, a simple approach to handling larger data sets and budgets may be to partition into smaller sub-problems (e.g., rather than selecting 5,000 points out of 50,000, instead partition to two sets of 25,000 and select 2,500 from each). We currently employ this in the classical active learning experiments (see Appendix E.5).
>
> **Features for greedy k-centers:** In each experiment, features for all baselines (including k-centers) are obtained in the same manner as our approach. This ensures a fair comparison. We recall that we compared active learning methods with different kinds of pretrained features:
> -	**Low Budget:** In Table 1 to 3 and Figure 2 & 4, all methods initialize with SimCLR features.
> -	**Domain Adaptation:** In Figure 3, all methods initialize with f-DAL features.
> -	**Classical:** In Table 4, all methods use features from VGG16 obtained from supervised learning in the previous round.
>
> **References**
>
> Altschuler et al., Massively Scalable Sinkhorn Distances via the Nystrom Method, NeurIPS 2019
>
> Xie et al., A fast proximal point method for computing exact wasserstein distance, UAI 2019

---

> > ### Comment · Reviewer_vpVq · 2021-11-29
> > **Appreciate the clarity.**
> >
> > Appreciate the clarity provided in the response. I retain my score-- I think the approach is interesting and novel. I would add that stronger uncertainty-based baselines (e.g. BALD and batchBALD) would be appreciated in the camera-ready (apologies for not raising this in my initial review), should the paper be accepted.

---

### Author Response · Authors · 2021-11-19
**Thank you to all the reviewers**

We thank all reviewers for their detailed comments and positive feedback on the paper. The main contribution of this paper is a new integer optimization framework for batch-mode deep active learning when the budget for labeling is restricted. This framework is supported by theoretical bounds and empirical results over different image classification data sets and tasks that demonstrate the value of optimization over conventional heuristics.

All reviewers remarked positively on the quality of writing and variety of experiments. We have revised the paper (with edits marked in red) to address reviewer feedback.

The main question was on the scalability of our method since SVHN is the largest dataset (73,257 images) we test our framework on. One approach to scale to larger datasets is to iteratively partition the given dataset into smaller datasets and thus reduce the selection problem into selection from smaller sub-problems.

Ultimately, the main bottleneck is computing Wasserstein distances. We used an exact approach to calculate them since we tested our approach on medium-sized datasets and our goal in this paper was to provide an analysis of our problem formulation. Nonetheless, massively scalable approximation techniques (e.g., Altschuler et al. 2019) can also be used. As this is an active field, we hope to explore new techniques for more efficiently computing these distances and further scaling our methods in future work.

**References**

Altschuler et al., Massively Scalable Sinkhorn Distances via the Nystrom Method, NeurIPS 2019

---

### Decision · Program_Chairs · 2022-01-20

**Decision:**

Accept (Poster)

**Comment:**

This is an interesting submission, which was overall well received by the reviewers. I would recommend the authors to discuss further the vast modern litterature on efficient computation of Wasserstein distances and their minimization (see, e.g. Peyré and Cuturi 2019, and references therein)